# FINITE-TIME BOUNDS FOR DISTRIBUTIONALLY RO-BUST TD LEARNING WITH LINEAR FUNCTION APPROX-IMATION

## ABSTRACT

Distributionally robust reinforcement learning (DRRL) focuses on designing policies that achieve good performance under model uncertainties. In particular, we are interested in maximizing the worst-case long-term discounted reward, where the data for RL comes from a nominal model while the deployed environment can deviate from the nominal model within a prescribed uncertainty set. Existing convergence guarantees for robust temporal-difference (TD) learning for policy evaluation are limited to tabular MDPs or are dependent on restrictive discount-factor assumptions when function approximation is used. We present the first robust TD learning with linear function approximation, where robustness is measured with respect to the total-variation distance uncertainty set. Additionally, our algorithm is both model-free and does not require generative access to the MDP. Our algorithm combines a two-time-scale stochastic-approximation update with an outer-loop target-network update. We establish an $\tilde{\mathcal{O}}(1/\epsilon^2)$ sample complexity to obtain an $\epsilon$-accurate value estimate. Our results close a key gap between the empirical success of robust RL algorithms and the non-asymptotic guarantees enjoyed by their non-robust counterparts. The key ideas in the paper also extend in a relatively straightforward fashion to robust Q-learning with function approximation.

## 1 INTRODUCTION

Reinforcement learning (RL) aims to learn policies that maximize long-term reward. Standard RL methods learn the optimal strategy from trajectories generated by a simulator or the real environment, implicitly assuming that training and deployment environments share the same dynamics. Many applications face two issues: simulation–reality gaps and distribution shift between training and deployment. These call for policies that are robust to perturbations in the environment. Distributionally robust RL (DRRL) tackles this by assuming the true environment lies in an uncertainty set around a nominal model. It then learns a policy that maximizes the worst-case cumulative reward over that set, using data from trajectories corresponding to the nominal model. In this work, we focus on model-free DRRL with linear function approximation for the value function to deal with large state spaces.

In contrast to our model-free approach, model-based DRRL often proceeds by fitting an empirical transition model, defining an uncertainty set from it, and then optimizing for a robust policy (Shi & Chi, 2024; Wang & Zou, 2021; Xu et al., 2023; Panaganti & Kalathil, 2022; Yang et al., 2022; Zhou et al., 2021). In some model-based papers, access to a generative-model is assumed, which is not realistic in many cases (Wang & Zou, 2021; Xu et al., 2023). Whether one assumes generative access or not, the number of parameters that need to be estimated in a model-based approach grows with the cardinality of the state and action spaces, unless one makes additional structural assumptions on the model.

Another line of work focuses on model-free learning of robust policies, that is, learning without constructing an empirical transition matrix. In the tabular setting, Liang et al. (2023) analyzes Cressie–Read $f$-divergence–based uncertainty sets and establishes asymptotic convergence guarantees for robust temporal-difference (TD) learning. A complementary tabular result, Li et al. (2022), studies the $R$-contamination uncertainty set and exploits a distinctive property: the robust Bellman operator

in this model admits an unbiased stochastic estimator. The techniques developed there extend to any uncertainty set that likewise permits an unbiased estimator of the robust Bellman operator, enabling unbiased policy evaluation and, consequently, policy improvement in a model-free manner. However, these papers do not consider function approximation, which is essential to deal with large state spaces.

When function approximation is introduced to represent the robust value function, the literature typically proceeds along two directions with different limitations. One line of research constructs the uncertainty set expressly so that the robust Bellman operator admits an unbiased estimator (Zhou et al., 2023), allowing standard stochastic approximation arguments to go through or restrict to $R$-contamination uncertainty set (Wang & Zou, 2021). For $R$-contamination uncertainty set, Wang & Zou (2021) investigates the TD-C algorithm under function approximation and provides finite-time bounds for convergence to a stationary point of the associated objective, offering non-asymptotic guarantees in a setting where the objective is nonconvex and only stationarity is generally attainable. The other direction assumes extremely small discount factors to induce a contraction mapping for the robust Bellman operator, which restores fixed-point uniqueness and enables convergence proofs Zhou et al. (2023); Badrinath & Kalathil (2021); Tamar et al. (2014). Both approaches trade generality for tractability: the first restricts attention to uncertainty sets with unbiased estimators and focuses only on local optimality, while the second relies on unrealistically small discounting to guarantee contraction.

Another line of work (Tang et al., 2024; Ma et al., 2022) for model-free DRRL considers linear Markov decision process (MDP) for DRRL where the transition matrix of the underlying MDP has a lower-dimensional structure. This reduces the complexity associated with large state spaces. In this paper, we do not make such a modeling assumption.

In summary, most existing results on model-free robust RL are limited in at least one crucial way: they prove only local or asymptotic convergence; focus on narrow uncertainty models (e.g., Liang et al. (2023) observe on FrozenLake that $R$-contamination–based methods can mirror non-robust baselines and even underperform due to over-conservatism); restricted to tabular settings; assume generative access; or require extremely small discount factors. In particular, there are no finite-time guarantees for robust TD with function approximation from a single trajectory under broad, practically motivated uncertainty classes—such as those induced by total variation or Wasserstein-$\ell$ distances. At the same time, practice-oriented deep-RL pipelines often use ad-hoc "robust TD" heuristics, leaving a sizable gap between theory and deployment. This work closes a portion of that gap by establishing finite-time guarantees for robust TD learning with function approximation under commonly used uncertainty sets, without relying on generative sampling, vanishing discount factors, or purely asymptotic arguments.

**Contributions.** Our main contributions are summarized below.

1. **Finite-time guarantees for Robust TD Learning** For total variation and Wasserstein-$\ell$ uncertainty sets, we establish that the distributionally robust policy evaluation considered in the paper with linear function approximation admits non-asymptotic guarantees from a single trajectory. The robust TD method achieves an $\epsilon$-accurate value estimate with sample complexity $\tilde{O}(1/\epsilon^2)$.

2. **Overcoming projection mismatch via target networks.** While the robust Bellman operator is a contraction in $\ell_\infty$ (Iyengar, 2005), function approximation induces a projected fixed-point equation that breaks direct contraction arguments. Prior approaches either remain tabular or require unrealistically small discount factors. We resolve this by incorporating a target-network mechanism—conceptually related to Munos & Szepesvári (2008) and, in the non-robust setting, Chen et al. (2023)—and prove stable, finite-time convergence of the resulting projected robust TD updates without restrictive discount-factor assumptions.

3. **Function approximation in the dual space.** Standard DRRL solvers compute the worst-case distribution at each step of an RL algorithm by using a dual formulation Iyengar (2005). However, this requires estimating a dual variable for each (state, action) pair, which is infeasible for large state spaces. To overcome this problem, we provide the first analysis of function approximation in the dual space.

4. **Robust Q-Learning.** The main technical contributions of the paper are in the proof of convergence and sample complexity bounds for robust TD learning with function approximation. It is straightforward to use these ideas to obtain finite-time bounds for robust Q-learning

with function approximation, which, to the best of our knowledge, has not been studied in the literature. We refer the reader to the short argument in the Appendix (Section E).

Since our paper focuses on discounted-reward robust RL, we have not made an exhaustive comparison of our work with work on average-reward robust RL; see, for example, Xu et al. (2025); Roch et al. (2025); Chen et al. (2025). However, to the best of our knowledge, it is worth noting that there are no performance guarantees even in the average-reward literature when function approximation is used.

## 2 MODEL AND PRELIMINARIES

**Model** We consider finite-state, finite-action, infinite-horizon discounted MDPs $\mathcal{M} := (\mathcal{S}, \mathcal{A}, P, R, \gamma)$, where $\mathcal{S}$ is the (finite) state space and $\mathcal{A}$ is the (finite) action space. For any finite set $\mathcal{X}$, we denote by $\Delta_{\mathcal{X}} := \left\{ \mu \in \mathbb{R}_+^{|\mathcal{X}|} : \sum_{x \in \mathcal{X}} \mu(x) = 1 \right\}$ the probability simplex over $\mathcal{X}$; in particular, $\Delta_{\mathcal{S}}$ and $\Delta_{\mathcal{A}}$ are the simplices over states and actions, respectively.

Throughout, we use lowercase letters $s \in \mathcal{S}$, $a \in \mathcal{A}$ to denote deterministic (non-random) states and actions, and uppercase letters $S, S', A$ to denote random states and actions taking values in $\mathcal{S}$ and $\mathcal{A}$. Given a state-action pair $(s, a)$, the transition kernel $P(\cdot \mid s, a) \in \Delta_{\mathcal{S}}$ specifies the distribution of the next-state random variable $S' \sim P(\cdot \mid s, a)$. The reward function is $R : \mathcal{S} \times \mathcal{A} \to [-1, 1]$, and $\gamma \in (0, 1)$ is the discount factor. A (stochastic) policy $\pi$ maps states to distributions over actions, that is, $\pi(\cdot \mid s) \in \Delta_{\mathcal{A}}$ for each $s \in \mathcal{S}$, and we write $\pi(a \mid s)$ for the probability of choosing action $a$ in state $s$.

Let $\{(S_t, A_t)\}_{t \geq 0}$ denote the state-action process for a policy $\pi$. Then for policy $\pi$ and transition model $P$, the (policy-dependent) state-action value is defined as

$$Q_P^{\pi}(s, a) := \mathbb{E}\left[ \sum_{t=0}^{\infty} \gamma^t R(S_t, A_t) \,\middle|\, S_0 = s, A_0 = a, \ A_t \sim \pi(\cdot \mid S_t), \ S_{t+1} \sim P(\cdot \mid S_t, A_t) \right].$$

**Robust MDPs (RMDPs) and uncertainty sets.** Distributionally robust RL (DRRL) models transition uncertainty via an *uncertainty set* around a nominal kernel $P_0$. We adopt the standard $(s, a)$-rectangular model (Iyengar, 2005; Nilim & El Ghaoui, 2005):

$$\mathcal{P}_s^a = \left\{ q \in \Delta_{\mathcal{S}} : D\big(q, P_0(\cdot \mid s, a)\big) \leq \delta \right\}, \qquad \mathcal{P} = \bigotimes_{(s,a) \in \mathcal{S} \times \mathcal{A}} \mathcal{P}_s^a, \tag{1}$$

where $D(\cdot, \cdot)$ is a probability distance or divergence (e.g., total variation or Wasserstein-$\ell$), and $\delta > 0$ is the radius. An RMDP is then the tuple (superscript 'rob' stands for "robust" throughout the rest of the paper)

$$\mathcal{M}^{\mathrm{rob}} = (\mathcal{S}, \mathcal{A}, \mathcal{P}, R, \gamma).$$

**Robust value functions (fixed policy).** Given a fixed policy $\pi$, the *robust* state-action value function is the worst-case value over $\mathcal{P}$:

$$Q^{\mathrm{rob},\pi}(s, a) := \min_{P \in \mathcal{P}} Q_P^{\pi}(s, a), \qquad V^{\mathrm{rob},\pi}(s) := \sum_a \pi(a \mid s) \, Q^{\mathrm{rob},\pi}(s, a). \tag{2}$$

It satisfies the *robust Bellman equation*:

$$Q^{\mathrm{rob},\pi}(s, a) = R(s, a) + \gamma \min_{q \in \mathcal{P}_s^a} \sum_{s'} q(s' \mid s, a) \underbrace{\sum_{a'} \pi(a' \mid s') \, Q^{\mathrm{rob},\pi}(s', a')}_{=: \ V^{\mathrm{rob},\pi}(s')}. \tag{3}$$

Equivalently, defining the robust Bellman operator $(\mathcal{T}^{\mathrm{rob},\pi} Q)(s, a) := R(s, a) + \gamma \, \sigma_{\mathcal{P}_s^a}(V^{Q,\pi}(s'))$ with

$$\sigma_{\mathcal{P}_s^a}(V) := \min_{q \in \mathcal{P}_s^a} \sum_{s'} q(s') \, V(s'), \qquad V^{Q,\pi}(s') := \sum_{a'} \pi(a' \mid s') \, Q(s', a'), \tag{4}$$

the fixed point relation is $Q^{\mathrm{rob},\pi} = \mathcal{T}^{\mathrm{rob},\pi} Q^{\mathrm{rob},\pi}$. We can write from the definitions,

$$0 \leq V^{\mathrm{rob},\pi}(s) \leq \frac{1}{1-\gamma}, \forall s \in \mathcal{S}; \qquad 0 \leq Q^{\mathrm{rob},\pi}(s, a) \leq \frac{1}{1-\gamma}, \forall (s, a) \in \mathcal{S} \times \mathcal{A}.$$

For a fixed $\pi$, evaluating $Q^{\mathrm{rob},\pi}$ reduces to solving Equation (3), which at each $(s,a)$ requires solving the inner problem Equation (4).

## 2.1 ROBUST TEMPORAL-DIFFERENCE LEARNING: CHALLENGES

**Function approximation.** Fix a policy $\pi$. We approximate the robust state-action value function by a linear function class with the learnable parameter vector $\theta \in \mathbb{R}^{n_\theta}$

$$Q_\theta^{\mathrm{rob},\pi}(s,a) \approx \phi(s,a)^\top \theta, \qquad \|\phi(s,a)\|_2 \leq 1, \forall (s,a) \in \mathcal{S} \times \mathcal{A}$$

with feature matrix $\Phi \in \mathbb{R}^{|\mathcal{S}||\mathcal{A}| \times n_\theta}$. Let $d^\pi(s,a)$ be the stationary state-action distribution of $(S_t, A_t)$ under $\pi$, and define $D^\pi := \mathrm{diag}\big(\{d^\pi(s,a)\}_{(s,a)\in\mathcal{S}\times\mathcal{A}}\big)$. Assume the weighted feature covariance is well-conditioned:

$$\Phi^\top D^\pi \Phi \succeq \mu I_{n_\theta} \quad \text{for some } \mu > 0.$$

Let $\mathcal{W} := \{\Phi\theta : \theta \in \mathbb{R}^{n_\theta}\}$ and denote by $\Pi : \mathbb{R}^{|\mathcal{S}||\mathcal{A}|} \to \mathcal{W}$ the $D^\pi$-orthogonal projection,

$$\Pi f = \Phi(\Phi^\top D^\pi \Phi)^{-1}\Phi^\top D^\pi f.$$

For any scalar $x \in \mathbb{R}$, we define the clipping operator

$$\mathrm{Clip}(x) := \min\Big\{\max\Big\{x, -\tfrac{1}{1-\gamma}\Big\}, \tfrac{1}{1-\gamma}\Big\}.$$

When applied to a vector $v \in \mathbb{R}^{n_\theta}$, $\mathrm{Clip}(v)$ denotes component-wise application of this operation.

We define the function approximation error for approximating the robust Q-function as:

$$\epsilon_{\mathrm{approx}} := \sup_{Q=\mathrm{Clip}(\Phi\theta),\theta\in\mathbb{R}^{n_\theta}} \big\|\mathrm{Clip}\big(\Pi\mathcal{T}^{\mathrm{rob},\pi}(Q)\big) - \mathcal{T}^{\mathrm{rob},\pi}(Q)\big\|_\infty. \tag{5}$$

**Key challenges in robust policy evaluation and our approach.** Model-free robust policy evaluation on a single trajectory typically hinges on a data-driven unbiased estimate $\hat{\sigma}_{\mathcal{P}_s^a}(V)$ of the inner-optimization objective defined in Equation (4). Except for special uncertainty sets (e.g., $R$-contamination), there is no direct plug-in *unbiased* single-sample estimator of this inner minimum, which creates a bias in standard TD updates. To overcome this challenge, we use a *two-time-scale* stochastic-approximation scheme in the inner loop of the algorithm: a fast time-scale solves for the inner-optimization problem defined in Equation (4) in its equivalent dual form, while the slow loop performs TD learning updates on $\theta$ using the estimate of the inner-optimization objective of the fast time-scale. Our two-time-scale algorithm is motivated by the algorithm in Liang et al. (2023), but the key difference here is the use of function approximation which necessitates a different analysis.

While $\mathcal{T}^{\mathrm{rob},\pi}$ is a $\gamma$-contraction in $\ell_\infty$-norm (Iyengar, 2005), function approximation introduces the *projected* operator $\Pi\mathcal{T}^{\mathrm{rob},\pi}$, which is *not* known to be a contraction in any norm for typical $\gamma \in (0,1)$. Prior work by Zhou et al. (2023) circumvents this by imposing restrictive assumptions on $\gamma$ which we do not adopt. We address the non-contraction of $\Pi\mathcal{T}^{\mathrm{rob},\pi}$ via a *target-network* mechanism prevalent in deep RL, analyzed by Munos & Szepesvári (2008) and later used in the non-robust setting by Chen et al. (2023), for Q-learning to overcome the contraction issue with the projected robust Bellman operator. At outer iteration $t$, we freeze a target parameter $\hat{\theta}_t$ and solve

$$\Phi\theta = \Pi\mathcal{T}^{\mathrm{rob},\pi}(\Phi\hat{\theta}_t)$$

in the inner loop, then update the target in the outer loop. This decoupling stabilizes the projected robust updates and enables our finite-time analysis under linear function approximation.

Standard DRRL literature solves the inner-optimization problem in Equation (4) in a corresponding dual space. However, solving it for each state-action pair is impractical for problems with large state and action spaces. We consider linear function approximation in the dual space of the optimization problem in Equation 4 and provide the first finite-sample analysis under this function approximation setup.

# 3 ROBUST TD LEARNING WITH LINEAR FUNCTION APPROXIMATION

## 3.1 UNCERTAINTY SETS

Before presenting the robust policy evaluation algorithm, we discuss the uncertainty sets considered in the paper: Total Variation (TV) uncertainty set and Wasserstein-$\ell$ uncertainty set.

**Total variation uncertainty set:** The total variation uncertainty set is defined as: for each $(s, a)$, $\mathcal{P}_s^{aTV} = \{q \in \Delta_\mathcal{S} : \frac{1}{2}\|q - P_0(\cdot|s,a)\|_1 \leq \delta\}$.

Simplifying (see: Appendix B) on the dual formulation originally given by Iyengar (2005) for the Total Variation uncertainty set, we get the following equivalent dual optimization:

$$\sigma_{\mathcal{P}_s^a}(V) \equiv \max_{\lambda_s^a \in [\frac{-1}{1-\gamma}, \frac{1}{1-\gamma}]} \{\mathbb{E}_{S \sim P_0(\cdot|s,a)}[\min(V(S), \lambda_s^a)] - \delta\lambda_s^a\}.$$

**Wasserstein-$\ell$ uncertainty set:** The uncertainty set is defined as: for each $(s, a)$: $\mathcal{P}_s^{aW_\ell} = \{q \in \Delta_\mathcal{S} : W_\ell(P_0(\cdot|s,a), q) \leq \delta\}$, where $\delta > 0$ is the uncertainty radius and $W_\ell(P_0(\cdot|s,a), q)$ is the Wasserstein-$\ell$ distance defined in detail in Appendix B.2.

The detailed analysis on TV and Wasserstein-$\ell$ uncertainty sets and the corresponding dual optimization problem are given in the Appendix B.

## 3.2 ALGORITHM AND MAIN RESULTS

In this subsection, we present our robust policy evaluation algorithm and the main results of the paper.

### 3.2.1 ROBUST POLICY EVALUATION ALGORITHM

Our robust TD learning algorithm is presented in Algorithm 1. In the rest of this section, we describe the algorithm and explain the notation used in the algorithm. In the outer loop (indexed by $t = 0, \cdots, T-1$), we freeze a *target parameter* $\hat{\theta}_t$; at the end of the inner loop we set $\hat{\theta}_{t+1}$ to the inner loop's final iterate. In the inner loop (indexed by $k = 0, \cdots, K-1$) we approximately solve for $\theta$ satisfying:

$$\Phi\theta = \Pi\mathcal{T}^{\mathrm{rob},\pi}(\Phi\hat{\theta}_t),$$

using a two-time-scale stochastic approximation: a fast loop for the dual variables corresponding to the inner-optimization problem 4, and a slow loop for the TD parameters. For a fixed outer loop $t$, the inner loop iterates are $\theta_{t,k}$ for $k \in [0, K-1]$.

At each inner loop iteration $k$, in a fast time-scale, we approximately solve the equivalent dual optimization problem in (4) using a super-gradient ascent step. Instead of maintaining a separate dual variable $\lambda_s^a$ for each $(s, a)$ (which would be tabular), we parameterize the dual variables $\lambda_s^a$ with the learnable parameter vector $\nu \in \mathbb{R}^{n_\lambda}$ as

$$\lambda_s^a \approx \psi(s, a)^\top \nu, \qquad \|\psi(s, a)\|_2 \leq 1, \forall (s, a) \in \mathcal{S} \times \mathcal{A},$$

with feature matrix $\Psi \in \mathbb{R}^{|\mathcal{S}||\mathcal{A}| \times n_\lambda}$.

Denote the robust value function estimate $V_{\hat{\theta}_t}^{\mathrm{rob}}$ evaluated at the target parameter $\hat{\theta}_t$ as

$$V_{\hat{\theta}_t}^{\mathrm{rob}}(s) = \sum_a \pi(a|s)\mathrm{Clip}\left(\phi(s, a)^\top\hat{\theta}_t\right), \forall s \in \mathcal{S}. \tag{6}$$

The quantity $V_{\hat{\theta}_t}^{\mathrm{rob}}$ can be computed exactly for any fixed target parameter $\hat{\theta}_t$. In the case of the TV distance uncertainty set, it suffices to compute $V_{\hat{\theta}_t}^{\mathrm{rob}}(s)$ *only for the state visited in each inner-loop iteration*, rather than for all states. We update $\nu_{t,k}$ with step-size $\beta_k$ using a projected *super-gradient ascent* on the dual objective with a super-gradient evaluated at the fresh data sample $(S_{t,k}, A_{t,k}, S_{t,k+1})$. Let $B_\nu > 0$ be a fixed finite radius, and define

$$\mathcal{M}_\nu := \left\{\nu \in \mathbb{R}^{n_\lambda} : \|\nu\|_2 \leq B_\nu\right\}.$$

The projection operator $\mathrm{Proj}_{\mathcal{M}_\nu}$ projects the dual parameter vector onto $\mathcal{M}_\nu$, ensuring that the iterates remain bounded. Since $\mathcal{M}_\nu$ is an $\ell_2$ ball, this projection can be computed by simple norm scaling.

In the algorithm, $\bar{\nu}_{t,k}$ denotes the half-tail iterate-average of the dual parameter vector, i.e.,

$$\bar{\nu}_{t,k} = \frac{1}{\lceil k/2 \rceil} \sum_{l=\lfloor k/2 \rfloor}^{k-1} \nu_{t,l} \tag{7}$$

which can be calculated easily by keeping track of the following two quantities: $\sum_{l=0}^{k-1} \nu_{t,l}$ and $\sum_{l=\lfloor k/2 \rfloor}^{k-1} \nu_{t,l}$. While many elements of our algorithm have been used in implementations of robust TD learning, to the best of our knowledge, such an averaging of the dual variables has not been used previously. The averaging turns out to be crucial in obtaining finite-time bounds, since it allows us to control the variance of the dual objective.

In the slow time-scale of the inner loop, $\theta_{t,k}$ is updated using asynchronous stochastic approximation with a step-size denoted by $\alpha_k$ with a robust TD-target $TD_{t,k+1}$. The two-time-scale scheme ensures that, at the slow scale, the dual variables appear near their sample-path equilibrium, yielding an (asymptotically) unbiased robust TD target.

---

**Algorithm 1** Robust TD learning with Function Approximation

---

1: **Input:** Integers $T, K$. Initial $\nu_0 \in \mathbb{R}^{n_\lambda}$, $\theta_0 :=$ zero vector, fast time-scale step-sizes $\beta_k = \frac{\beta_0}{\sqrt{k+1}}$, for some $0 < \beta_0 < \infty$, slow time-scale step-sizes $\alpha_k = \frac{c}{(k+1)}$ for some $0 < c < \infty$; $\hat{\theta}_0 = \theta_0$, $\theta_{0,0} = \theta_0$, candidate policy $\pi$, Reward function $R : (\mathcal{S} \times \mathcal{A}) \mapsto [-1,1]$, initial state $S_{0,0}$.
2: **for** $t = 0, 1, \ldots, T-1$ **do**
3:     **for** $k = 0, 1, \ldots, K-1$ **do**
4:         Take action $A_{t,k}$ according to policy $\pi$ and sample $S_{t,k+1}$ ($S_{t,k+1} \sim P_0(\cdot|S_{t,k}, A_{t,k})$)
5:     **fast time-scale** $(\beta_k)$
6:         Compute $\hat{G}(\psi(S_{t,k}, A_{t,k})^\top \nu_{t,k}; V_{\hat{\theta}_t}^{\mathrm{rob}}, S_{t,k+1})$ from Equation (17) for TV uncertainty set and Equation (20) for Wasserstein-$\ell$ uncertainty set
7:         $\nu_{t,k+1} = \mathrm{Proj}_{\mathcal{M}_\nu}(\nu_{t,k} + \beta_k[\hat{G}(\psi(S_{t,k}, A_{t,k})^\top \nu_{t,k}; V_{\hat{\theta}_t}^{\mathrm{rob}}, S_{t,k+1})\psi(S_{t,k}, A_{t,k})])$
8:     **Slow scale** $(\alpha_k)$
9:         Compute $\bar{\nu}_{t,k}$ from Equation (7)
10:        Compute $\hat{F}(\psi(S_{t,k}, A_{t,k})^\top \bar{\nu}_{t,k}; V_{\hat{\theta}_t}^{\mathrm{rob}}, S_{t,k+1})$ from Equation (18) for TV uncertainty set and Equation (21) for Wasserstein-$\ell$ uncertainty set
11:        $TD_{t,k+1} = R(S_{t,k}, A_{t,k}) + \gamma \hat{F}(\psi(S_{t,k}, A_{t,k})^\top \bar{\nu}_{t,k}; V_{\hat{\theta}_t}^{\mathrm{rob}}, S_{t,k+1}) - \phi(S_{t,k}, A_{t,k})^\top \theta_{t,k}$
12:        $\theta_{t,k+1} = \theta_{t,k} + \alpha_k TD_{t,k+1}\phi(S_{t,k}, A_{t,k})$
13:     **end for**
14:     $\hat{\theta}_{t+1} = \theta_{t,K}$, $S_{t+1,0} = S_{t,K}$, $\theta_{t+1,0} = \theta_{t,K}$, $\nu_{t+1,0} = \nu_{t,K}$.
15: **end for**
16: **Output:** $\hat{\theta}_T$

---

### 3.2.2 MAIN RESULT

We define the function approximation error for approximating the dual variables next. For compactness of notation, denote for a value function $V$, for each $(s,a)$, $F_{s,a}^{*,V} := \sup_{\lambda_s^a} F(\lambda_s^a; V, P_0(\cdot|s,a))$ and $F^V(\nu)_{s,a} := F(\psi(s,a)^\top \nu; V, P_0(\cdot|s,a))$. Define

$$\epsilon_{\mathrm{approx}}^{\mathrm{dual}} := \sup_{V:V(s)=\sum_a \pi(a|s)\mathrm{Clip}(\phi(s,a)^\top \theta); \theta \in \mathbb{R}^{n_\theta}} \inf_{\nu \in \mathcal{M}_\nu} \|F^{*,V} - F^V(\nu)\|_\infty. \tag{8}$$

We make the following assumption on the policy $\pi$.

**Assumption 1.** *The policy $\pi$ induces an irreducible and aperiodic Markov chain on $\mathcal{S} \times \mathcal{A}$ under the nominal transition kernel $P_0$.*

For $\tau \leq k$, define

$$\eta_k^{t,\tau}(\cdot) := \mathbb{P}\big((S_{t,k}, A_{t,k}) \in \cdot \,\big|\, S_{t,0}, A_{t,0}, \ldots, S_{t,k-\tau}, A_{t,k-\tau}\big).$$

Under Assumption 1, the Markov chain is geometrically mixing: there exist constants $C_{\text{mix}} < \infty$ and $\rho \in (0,1)$ such that

$$\big\|\eta_k^{t,\tau} - d^\pi\big\|_{\text{TV}} \leq C_{\text{mix}} \, \rho^\tau, \qquad \forall t, k, \tau.$$

Here $C_{\text{mix}}$ and $\rho$ depend only on the nominal model $(P_0, \pi)$.

Let $\hat{Q}_t := \text{Clip}\left(\Phi\hat{\theta}_t\right)$ be the estimate of $Q^{\text{rob},\pi}$ by Algorithm 1 at outer iteration $t$.

In Theorem 1, we present our main result, which establishes the convergence of $\hat{Q}_T$ to the robust value function $Q^{\text{rob},\pi}$, up to terms arising from function-approximation error.

**Theorem 1** (Finite-time bound: rates and dependencies (informal))**.** *Define*

$$k_{\text{mix}}$$
$$:= \min\left\{ m \in \mathbb{N} : \forall j \geq m, \; j \; \geq \; \max\left(\tau_\mu, 2\Big\lceil \frac{\log\big(\frac{C_{\text{mix}}}{\beta_0}\sqrt{j+1}\big)}{\log(1/\rho)}\Big\rceil, \; \Big\lceil \frac{\log\big(C_{\text{mix}}(j+1)/c\big)}{\log(1/\rho)}\Big\rceil\right)\right\},$$

*where*

$$\tau_\mu := \Big\lceil \frac{\log\big(C_{\text{mix}}\frac{1}{\mu}\big)}{\log(1/\rho)}\Big\rceil.$$

*Assume Assumption 1 holds, and we run $K \geq k_{\text{mix}}$ inner iterations per outer iteration for either the TV uncertainty set or the Wasserstein-$\ell$ uncertainty set. Then, for any $T \geq 1$, we have*

$$\mathbb{E}\big[\|\hat{Q}_T - Q^{\text{rob},\pi}\|_\infty\big]$$

$$\leq \gamma^T \, \|\text{Clip}(\Phi\theta_0) - Q^{\text{rob},\pi}\|_\infty \; + \; \frac{\text{rate}_{\text{inner}}(K)}{(1-\gamma)^2} \; + \; \frac{\epsilon_{\text{approx}}}{1-\gamma} \; + \; \frac{2\sqrt{2}\left(1 + \frac{2}{K}\right)^{\frac{\mu c}{4}} \epsilon_{\text{approx}}^{\text{dual}}}{\mu(1-\gamma)},$$

*where the term $\text{rate}_{\text{inner}}(K)$ is of the following order in terms of inner iteration number $K$:*

$$\text{rate}_{\text{inner}}(K) = \begin{cases} \mathcal{O}\big(K^{-\mu c/4}\big), & \alpha_k = \dfrac{c}{k+1}, \quad \mu c < 2, \\[2mm] \mathcal{O}\big((\log K)^{1/2}K^{-1/2}\big), & \alpha_k = \dfrac{c}{k+1}, \quad \mu c = 2, \\[2mm] \mathcal{O}\big(K^{-1/2}\big), & \alpha_k = \dfrac{c}{k+1}, \quad \mu c > 2, \end{cases}$$

*where the notation $\mathcal{O}$ captures the problem-dependent constants depending on $(\mu, \delta, C_{\text{mix}}, \rho, B_\nu, \beta_0, c)$.*

**Remark 1.** *A fully constant-explicit version of Theorem 1 is provided in Theorem 2 in the Appendix.*

Recall the slow time-scale step-size rule is $\alpha_k = \frac{c}{k+1}, \forall k$. The sample complexity to achieve an $\epsilon$-approximate robust Q-function estimate can be derived in the following manner. Assume $\mu c > 2$. If we choose $T = \mathcal{O}\left(\ln\left(\frac{1}{\epsilon(1-\gamma)}\right)\right)$ and $K = \mathcal{O}\left(\frac{1}{(\epsilon(1-\gamma)^2)^2}\right)$, we have $\gamma^T \|\text{Clip}(\Phi\theta_0) - Q^{\text{rob},\pi}\|_\infty + \frac{\text{rate}_{\text{inner}}(K)}{(1-\gamma)^2} = \mathcal{O}(\epsilon)$. This gives us the following sample complexity result.

**Corollary 1** (Sample Complexity)**.** *Suppose the step-size rule $\alpha_k = \frac{c}{1+k}$ is used with $\mu c \geq 2$. Then the sample complexity for Algorithm 1 achieves an element-wise $\epsilon$-accurate estimate of $Q^{\text{rob},\pi}$ up to the function approximation error is*

$$\mathcal{O}\left(\ln\left(\frac{1}{\epsilon(1-\gamma)}\right) \frac{1}{\epsilon^2(1-\gamma)^4}\right). \tag{9}$$

*Similar sample complexity results can be obtained for other values of $\mu c$.*

We note that the step-size rule $\frac{c}{k+1}$ achieves the best sample complexity, but it requires $c$ to be chosen sufficiently large. This is consistent with similar results in the non-robust RL literature; see, for example, Chen et al. (2023).

## 4   KEY IDEAS AND PROOF OUTLINE

While the detailed proof of Theorem 1 is presented in Appendix C, we provide the key ideas behind the proof in this section.

We define the stacked reward vector $r \in \mathbb{R}^{|\mathcal{S}||\mathcal{A}|}$ by

$$r_{s,a} := R(s,a), \qquad (s,a) \in \mathcal{S} \times \mathcal{A},$$

using some fixed ordering of state-action pairs.

Fix an outer loop iteration $t$. Recall the definition $F^{*,V}_{s,a} := \sup_{\lambda^a_s} F(\lambda^a_s; V, P_0(\cdot|s,a))$. Define the inner loop error for outer iteration index $t$ as $e_{t,k} := \theta_{t,k} - \theta^*_t$ with

$$\theta^*_t := (\Phi^\top D^\pi \Phi)^{-1} \Phi^\top D^\pi \big[ r + \gamma\, F^{*,V^{\mathrm{rob}}_{\hat{\theta}_t}} \big]. \tag{10}$$

The next lemma bounds the expected estimation error at the final outer-loop iterate in terms of the inner-loop error terms.

**Lemma 1.** *Under the setting in Theorem 1, Algorithm 1 guarantees*

$$\mathbb{E}\left[\|\hat{Q}_T - Q^{\mathrm{rob},\pi}\|_\infty\right] \le \gamma^T \|\mathrm{Clip}\,(\Phi\theta_0) - Q^{\mathrm{rob},\pi}\|_\infty + \underbrace{\sum_{t=1}^{T} \gamma^{T-t} \mathbb{E}\left[\|e_{t,K}\|_\infty\right]}_{\textit{Inner loop convergence error}} + \frac{\epsilon_{\mathrm{approx}}}{1-\gamma}.$$

The proof of Lemma 1 is provided in Appendix C.1 and is inspired by the analysis in Chen et al. (2023) for non-robust Q-learning.

In the analysis that follows, we establish that the inner loop error remains small (up to function approximation error terms) in $\ell_\infty$-norm for sufficiently large $k$. We decompose the slow time-scale update at inner loop $k$ in Algorithm 1 into mean drift, noise and bias terms as

$$\theta_{t,k+1} = \theta_{t,k} + \alpha_k \left[ H(\theta_{t,k}) \; + \; b^\theta_{t,k} \; + \; n^\theta_{t,k+1} \right],$$

where

$$H(\theta_{t,k}) := \Phi^\top D^\pi \big[ r + \gamma F^{*,V^{\mathrm{rob}}_{\hat{\theta}_t}} - \Phi\theta_{t,k} \big] \underbrace{=}_{\text{from Equation (10)}} \Phi^\top D^\pi \Phi(\theta^*_t - \theta_{t,k}),$$

$$b^\theta_{t,k} := \gamma \Phi^\top D^\pi \left[ F^{V^{\mathrm{rob}}_{\hat{\theta}_t}}(\bar{\nu}_{t,k}) - F^{*,V^{\mathrm{rob}}_{\hat{\theta}_t}} \right],$$

$$n^\theta_{t,k+1} := TD_{t,k+1}\, \phi(S_{t,k}, A_{t,k}) - H(\theta_{t,k}) - b^\theta_{t,k}.$$

**Idealized recursion (without noise and bias).**   The mean drift term corresponds to the deterministic recursion:

$$\theta_{t,k+1} = \theta_{t,k} + \alpha_k \Phi^\top D^\pi \Phi(\theta^*_t - \theta_{t,k}).$$

This recursion admits $\theta^*_t$ as its unique fixed point. Since the matrix $\Phi^\top D^\pi \Phi$ is symmetric and positive definite with minimum eigenvalue $\mu > 0$, in the absence of bias and noise terms, the iterates satisfy

$$\|\theta_{t,k+1} - \theta^*_t\|_2 \; \le \; (1 - \alpha_k \mu)\, \|\theta_{t,k} - \theta^*_t\|_2, \tag{11}$$

which implies geometric convergence of $\theta_{t,k}$ to $\theta^*_t$ at a rate governed by $\mu$.

**Bias term analysis.**   Recall that the bias term is given by

$$b^\theta_{t,k} \; := \; \gamma \Phi^\top D^\pi \left[ F^{V^{\mathrm{rob}}_{\hat{\theta}_t}}(\bar{\nu}_{t,k}) - F^{*,V^{\mathrm{rob}}_{\hat{\theta}_t}} \right].$$

We show that this term becomes small for large $k$, up to a function approximation error $\epsilon^{\mathrm{dual}}_{\mathrm{approx}}$.

In the fast time-scale analysis, we prove that the stochastic update on $\nu$ performs a super-gradient ascent on the concave objective

$$L_t(\nu) \; := \; \sum_{s,a} d^\pi(s,a)\, F\big(\psi(s,a)^\top \nu;\, V^{\mathrm{rob}}_{\hat{\theta}_t}, P_0(\cdot|s,a)\big),$$

which has bounded super-gradients. By a standard Lyapunov function argument for stochastic approximation under a mixing Markov chain, we obtain the following guarantee on the iterates from the fast time-scale for sufficiently large $k$ (stated in detail in Lemma 4 in Appendix C):

$$\mathbb{E}\left[\max_{\nu \in \mathcal{M}_\nu} L_t(\nu) - L_t(\bar{\nu}_{t,k})\right] \leq \frac{C_{\text{fast}}}{\sqrt{k}}, \tag{12}$$

where the constant $C_{\text{fast}}$ is given in equation 25.

Using $\|\phi(s,a)\|_2 \leq 1$ and $F_{s,a}^{*,V_{\hat{\theta}_t}^{\text{rob}}} \geq F^{V_{\hat{\theta}_t}^{\text{rob}}}(\bar{\nu}_{t,k})_{s,a}$ for all $(s,a)$, we can write

$$\|b_{t,k}^\theta\|_2 \leq \gamma \sum_{s,a} d^\pi(s,a) \left| F^{V_{\hat{\theta}_t}^{\text{rob}}}(\bar{\nu}_{t,k})_{s,a} - F_{s,a}^{*,V_{\hat{\theta}_t}^{\text{rob}}} \right| = \gamma \sum_{s,a} d^\pi(s,a) \left( F_{s,a}^{*,V_{\hat{\theta}_t}^{\text{rob}}} - F^{V_{\hat{\theta}_t}^{\text{rob}}}(\bar{\nu}_{t,k})_{s,a} \right)$$

$$\leq \gamma \underbrace{\inf_{\nu \in \mathcal{M}_\nu} \sum_{s,a} d^\pi(s,a) \left( F_{s,a}^{*,V_{\hat{\theta}_t}^{\text{rob}}} - F^{V_{\hat{\theta}_t}^{\text{rob}}}(\nu)_{s,a} \right)}_{\leq \epsilon_{\text{approx}}^{\text{dual}}} + \gamma \underbrace{\left[ \sup_{\nu \in \mathcal{M}_\nu} L_t(\nu) - L_t(\bar{\nu}_{t,k}) \right]}_{\text{fast-scale objective gap}}.$$

**Handling the noise term.** Finally, to handle the noise terms $n_{t,k+1}^\theta$, we employ the approach in Srikant & Ying (2019), where a bound is obtained on the expectation of the error $\|\theta_{t,k} - \theta_t^*\|_2^2$ conditioned with respect to the filtration generated by the set $(S_{t,0}, A_{t,0}, S_{t,1}, A_{t,1}, ..., S_{t,k-\tau}, A_{t,k-\tau})$. By choosing a lag $\tau$ such that the underlying Markov chain has mixed sufficiently, the effect of noise can be controlled.

## 5 DISCUSSION

As mentioned in the introduction, we provide the first proof of convergence and finite-time bounds for robust TD learning with function approximation, without making any assumptions on the underlying model or imposing very restrictive assumptions on the discount factor. Some immediate extensions and open problems are identified below:

1. The algorithm and the results can be extended to other families of distances between probability distributions, such as the Cressie-Read family of $f$-divergences considered in Liang et al. (2023), which admit duality representations that allow one to obtain unbiased estimators of the quantities of interest. For the Cressie-Read family, this would require the addition of one more time-scale, but the rest of the analysis would be similar. Our results also apply to the $R$-contamination set, but the algorithm is even simpler in that case due to the fact that the dual problem has a closed-form solution Xu et al. (2025).

2. Although the results in the main body of the paper have been presented for robust TD learning, they can be easily extended to robust Q-learning with function approximation to obtain optimal policies; see the Appendix.

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

# A  CONTENTS

The contents of the Appendix are as follows:

1. In Section B, we analyze the TV distance and Wasserstein-$\ell$ uncertainty sets in detail.

2. Section C proves the main result of the paper, that is, Theorem 1 in detail.

3. In Section E, we present the robust Q learning algorithm with linear function approximation (Algorithm 2) and discuss how the theoretical analysis for robust TD learning can be extended to the robust Q learning straightforwardly.

# B  CONDITIONS FOLLOWED BY TV AND WASSERSTEIN-$\ell$ UNCERTAINTY SETS

To aid the analysis, in the this section we outline a few properties of the uncertainty sets considered in this paper: TV and Wasserstein-$\ell$ uncertainty sets. In Section 5, we discuss how our algorithm can be trivially modified to satisfy a similar convergence guarantee for the R-contamination uncertainty set and Cressie-Read family of f-divergences considered in Liang et al. (2023).

**Lemma 2.** *The TV and Wasserstein uncertainty sets considered in this paper satisfies the following conditions. The optimization problem $\sigma_{\mathcal{P}_s^a}(V)$ for a generic value function $V$ as defined in Equation (4) has an equivalent dual optimization problem corresponding to a dual variable $\lambda_s^a$ :*

$$\sigma_{\mathcal{P}_s^a}(V) \equiv \sup_{\lambda_s^a \geq 0} \left( F(\lambda_s^a; V, P_0(\cdot|s,a)) \right)$$

*where $F(\lambda_s^a; V, P_0(\cdot|s,a))$ is a $\lambda_s^a$-concave function with the following properties:*

1. *Let $G(\lambda_s^a; V, P_0(\cdot \mid s,a))$ be a super-gradient of the concave function $F(\lambda_s^a; V, P_0(\cdot \mid s,a))$. There exists an unbiased estimator $\hat{G}(\lambda_s^a; V, S')$ of $G(\lambda_s^a; V, P_0(\cdot \mid s,a))$ based on a sample of the next state $S' \sim P_0(\cdot \mid s,a)$, that is,*

$$\mathbb{E}_{S' \sim P_0(\cdot|s,a)}[\hat{G}(\lambda_s^a; V, S')] = G(\lambda_s^a; V, P_0(\cdot \mid s,a)),$$

*and it satisfies $|\hat{G}(\lambda_s^a; V, S')| \leq C_G < \infty$ for all $\lambda_s^a \in \mathbb{R}$ for some constant $C_G \geq 0$.*

2. *There exists an unbiased estimator $\hat{F}(\lambda_s^a; V, S')$ of the dual objective $F(\lambda_s^a; V, P_0(\cdot|s,a))$ based on a sample of next state $S' \sim P_0(\cdot|s,a)$, that is,*

$$\mathbb{E}_{S' \sim P_0(\cdot|s,a)}[\hat{F}(\lambda_s^a; V, S')] = F(\lambda_s^a; V, P_0(\cdot|s,a)).$$

*Moreover, the estimator is uniformly bounded on bounded sets of $\lambda_s^a$ : for every $M > 0$ there exists a constant $C_{F,M} < \infty$ such that, for all $|\lambda_s^a| \leq M$ and all $s' \in \mathcal{S}$,*

$$\left| \hat{F}(\lambda_s^a; V, s') \right| \leq C_{F,M}.$$

Next, we discuss in detail the uncertainty sets considered in this paper, namely, TV distance uncertainty and Wasserstein-$\ell$ uncertainty sets and prove the Lemma 2. For each uncertainty set,

1. We define the uncertainty set first. Then, we discuss and analyze the equivalent dual optimization that corresponds to the inner-optimization problem defined in Equation 4.

2. We show the uncertainty set satisfies the conditions described in Lemma 2 and hence prove Lemma 2.

## B.1  TOTAL VARIATION DISTANCE UNCERTAINTY SET

The total variation uncertainty set is defined for each $(s, a)$ pair as,

$$\mathcal{P}_s^{aTV} = \{q \in \Delta_{\mathcal{S}} : \frac{1}{2}\|q - P_0(\cdot|s,a)\|_1 \leq \delta\}.$$

Next, we show that the optimization problem given in Equation 4 in the main body of the paper with $\mathcal{P}_s^{a\,TV}$ as the uncertainty set satisfies the conditions described in Lemma 2. Let us rewrite the optimization problem here for the TV distance uncertainty set.

$$\sigma_{\mathcal{P}_s^{a\,TV}}(V) = \min_{q \in \mathcal{P}_s^{a\,TV}} q^\top V.$$

From Lemma 4.3 in Iyengar (2005), we know that the above optimization problem can be solved under the dual formulation :

$$\sigma_{\mathcal{P}_s^{a\,TV}}(V) = \max_{f \in \mathbb{R}_+^{|\mathcal{S}|}} \left( \mathbb{E}_{S \sim P_0(\cdot|s,a)}[V(S) - f(S)] - \delta \, span(V - f) \right). \tag{13}$$

Next, we prove that the above dual optimization problem is equivalent to a scalar optimization problem.

**Lemma 3.** *The optimization problem given in Equation (13) is equivalent to the following optimization problem:*

$$\sigma_{\mathcal{P}_s^{a,TV}}(V) \equiv \delta \min_{s'} V(s') + \max_{\lambda_s^a \in [\min_{s'} V(s'), \max_{s'} V(s')]} \{\mathbb{E}_{S \sim P_0(\cdot|s,a)}[\min\{V(S), \lambda_s^a\}] - \delta \lambda_s^a\}. \tag{14}$$

*Proof.* **From the $\mu$-vector dual to a 1–D cut off problem:** The optimization problem in Equation (13) can be written as

$$\max_{f \in \mathbb{R}_+^{|\mathcal{S}|}} \left\{ \mathbb{E}_{S \sim P_0(\cdot|s,a)}[V(S) - f(S)] - \delta \left[ \max_{s'}(V(s') - f(s')) - \min_{s'}(V(s') - f(s')) \right] \right\}. \tag{15}$$

**Step 1 – restrict to "cut–off" vectors:** For any scalar $z \in [\min_{s'} V(s'), \max_{s'} V(s')]$, define

$$f_z(s) := \big[V(s) - z\big]_+ = \max\{0, \, V(s) - z\}.$$

Replacing an arbitrary feasible $f$ by the corresponding $f_{z:=\max_{s'}(V(s')-f(s'))}$ cannot decrease the objective in equation 15, so an optimizer always has the form $f_{z^*}$ for some $z^* \in [\min_{s'} V(s'), \max_{s'} V(s')]$.

**Step 2 – plug $f_z$ into the objective.** Because $V(s) - f_z(s) = \min\{V(s), z\}$,

$$\max_s(V - f_z) = z, \qquad \min_s(V - f_z) = \min_{s'} V(s'),$$

and

$$\mathbb{E}_{S \sim P_0}[V(S) - f_z(S)] = \mathbb{E}_{S \sim P_0}\big[\min\{V(S), z\}\big].$$

Substituting these identities into equation 15 yields the *scalar* optimization

$$\sigma_{\mathcal{P}_s^{a,TV}}(V) = \delta \min_{s'} V(s') + \max_{z \in [\min_{s'} V(s'), \max_{s'} V(s')]} \left\{ \mathbb{E}_{S \sim P_0(\cdot|s,a)}\big[\min\{V(S), z\}\big] - \delta z \right\}. \tag{16}$$

$\square$

As we are dealing with $V$ functions for which $V(s) \in \{\frac{-1}{1-\gamma}, \frac{1}{1-\gamma}\}$, the optimum dual variable lies in: $\lambda_s^a \in \{\frac{-1}{1-\gamma}, \frac{1}{1-\gamma}\}$ and we can equivalently write from Lemma 3,

$$\sigma_{\mathcal{P}_s^{a,TV}}(V) = \delta \min_{s'} V(s') + \max_{\lambda_s^a \in \{\frac{-1}{1-\gamma}, \frac{1}{1-\gamma}\}} \{\mathbb{E}_{S \sim P_0(\cdot|s,a)}[\min V(S), \lambda_s^a] - \delta \lambda_s^a\}.$$

It is easy to verify that the concave objective has a super-gradient:

$$G^{TV}(\lambda_s^a; V, P_0(\cdot|s,a)) := \mathbb{P}_{S \sim P_0(\cdot|s,a)}[V(S) \geq \lambda_s^a] - \delta.$$

An unbiased estimate of the super-gradient for a value of $\lambda_s^a$ and the value function $V$ from a next state $S' \sim P_0(\cdot|s,a)$ can be given as:

$$\hat{G}^{TV}(\lambda_s^a; V, S') := \mathbf{1}_{V(S') \geq \lambda_s^a} - \delta. \tag{17}$$

An unbiased estimate of the dual objective for a value of $\lambda_s^a$ and the value function $V$ from a next state $S' \sim P_0(\cdot|s,a)$ can be given as

$$\hat{F}^{TV}(\lambda_s^a; V, S') = \delta \min_{s'} V(s') + \min(V(S'), \lambda_s^a) - \delta\lambda_s^a. \tag{18}$$

As we have $|V(s)| \leq \frac{1}{1-\gamma}, \forall s \in \mathcal{S}$, its easy to see that,

$$|\hat{G}^{TV}(\lambda_s^a; V, S')| \leq C_G^{TV} := \max(\delta, 1-\delta), \forall \lambda_s^a \in \mathbb{R},$$

and, for any $0 < M < \infty$,

$$|\hat{F}^{TV}(\lambda_s^a; V, S')| \leq C_{F,M}^{TV} := (1+\delta)\left(M + \frac{1}{1-\gamma}\right), \forall \lambda_s^a \in [-M, M].$$

### B.2 WASSERSTEIN-$\ell$ UNCERTAINTY SET

We define the Wasserstein-$\ell$ uncertainty set for each $(s,a)$ pair as:

$$\mathcal{P}_s^{a\,W_\ell} = \{q \in \Delta_\mathcal{S} : W_\ell(P_0(\cdot|s,a), q) \leq \delta\},$$

where $\delta > 0$ is the uncertainty radius and $W_\ell(P_0(\cdot|s,a), q)$ is the Wasserstein-$\ell$ distance defined next. Consider the generic metric space $(\mathcal{S}, d)$ by defining some distance metric $d$. For some parameter $\ell \in [1, \infty)$, and two distributions $p, q \in \Delta_\mathcal{S}$, define the Wasserstein-$\ell$ distance between them as $W_\ell(q, p) = \inf_{N \in \Gamma(p,q)} \|d\|_{N,\ell}$, where $\Gamma(p, q)$ denotes the distribution over $\mathcal{S} \times \mathcal{S}$ with marginal distributions $p, q$ and $\|d\|_{N,\ell} = (\mathbb{E}_{(X,Y) \sim N}[d(X,Y)^\ell])^{1/\ell}$. Let us use the distance matrix with normalization, ensuring $|d(s, s')| \leq 1, \forall (s, s')$.

Next, we show that the following optimization problem with $\mathcal{P}_s^{a\,W_\ell}$ as the uncertainty set satisfies the conditions described in Lemma 2.

$$\sigma_{\mathcal{P}_s^{a\,W_\ell}}(V) = \min_{q \in \mathcal{P}_s^{a\,W_\ell}} q^\top V.$$

From Gao & Kleywegt (2023), we know that the above optimization problem can be solved under the dual formulation :

$$\sigma_{\mathcal{P}_s^a}(V) = \sup_{\lambda_s^a \geq 0}\left(-\lambda_s^a \delta^\ell + \mathbb{E}_{P_0(\cdot|s,a)}[\inf_y(V(y) + \lambda_s^a d(S, y)^\ell)]\right).$$

As the state space $\mathcal{S}$ is finite, we can replace the inner-optimization $[\inf_y(V(y) + \lambda_s^a d(S, y)^\ell)]$ with $[\min_y(V(y) + \lambda_s^a d(S, y)^\ell)]$. Next, we show that the optimum dual variable of the above optimization problem lies inside a compact set $\left[0, \lambda_M^{W_\ell}\right]$ with $\lambda_M^{W_\ell} := \frac{span(V)}{\delta^\ell}$.

As point-wise minimum of affine functions is concave, the above optimization problem is a concave optimization problem. It is easy to verify that the concave objective has a super-gradient:

$$G^{W_\ell}(\lambda_s^a; V, P_0(\cdot|s,a)) = -\delta^\ell + \mathbb{E}_{X \sim P_0(\cdot|s,a)}[d(X, y_{\lambda_s^a}^*(X))^\ell], \tag{19}$$

where,

$$y_{\lambda_s^a}^*(x) \in \arg\min_y[V(y) + \lambda_s^a d(x, y)^\ell].$$

Let us fix an $S = s$ and its minimizer $y_{\lambda_s^a}^*(x)$ for the inner-optimization $[\inf_y(V(y) + \lambda_s^a d(s, y)^\ell)]$. Because the candidate $y = s$ is always feasible,

$$V(y_{\lambda_s^a}^*(s)) + \lambda_s^a d(s, y_{\lambda_s^a}^*)^\ell \leq V(s).$$

Rearrange:

$$d(s, y_{\lambda_s^a}^*(s))^\ell \leq \frac{V(s) - V(y_{\lambda_s^a}^*(s))}{\lambda_s^a} \leq \frac{span(V)}{\lambda_s^a}.$$

Taking expectation in Equation 19 and using the above equation gives

$$G^{W_\ell}(\lambda_s^a; V, P_0(\cdot|s,a)) \leq -\delta^\ell + \frac{span(V)}{\lambda_s^a}.$$

Now, for any $\lambda_s^a > \lambda_M^{W_\ell} = \frac{span(V)}{\delta^\ell}$, we have,

$$G^{W_\ell}(\lambda_s^a; V, P_0(\cdot|s,a)) \leq 0.$$

Due to the concavity of the objective, a non-positive super-gradient means the function is non-increasing for all $\lambda_s^a > \lambda_M^{W_\ell}$. Combining the observation with the boundedness of the objective for bounded $\lambda_s^a$, we conclude that the supremum is attained and lies in $[0, \lambda_M^{W_\ell}]$.

An unbiased estimate of the super-gradient for a value of $\lambda_s^a$ and the value function $V$ from a next state $S' \sim P_0(\cdot|s,a)$ can be given as:

$$\hat{G}^{W_\ell}(\lambda_s^a; V, S') = -\delta^\ell + d(S', y^{*'})^\ell, \tag{20}$$

where,

$$y^{*'} = \arg\min_y [V(y) + \lambda_s^a d(S', y)^\ell].$$

An unbiased estimate of the dual objective for a value of $\lambda_s^a$ and the value function $V$ from a next state $S' \sim P_0(\cdot|s,a)$ can be given as

$$\hat{F}^{W_\ell}(\lambda_s^a; V, S') = -\lambda_s^a \delta^\ell + V(y^{*'}) + \lambda_s^a d(S', y^{*'})^\ell. \tag{21}$$

If we assume $|V(s)| \leq \frac{1}{1-\gamma}, \forall s \in \mathcal{S}$, its easy to show that,

$$|\hat{G}^{W_\ell}(\lambda_s^a; V, S')| \leq C_G^{W_\ell} := 1 + \delta^\ell, \forall \lambda_s^a \in \mathbb{R},$$

and, for any $0 < M < \infty$,

$$|\hat{F}^{W_\ell}(\lambda_s^a; V, S')| \leq C_{F,M}^{W_\ell} := (\delta^\ell + 1)M + \frac{1}{1-\gamma}, \forall \lambda_s^a \in [-M, M].$$

## C   CONVERGENCE ANALYSIS OF ALGORITHM 1 AND THE PROOF OF THEOREM 1

In this section, we provide the proof of Theorem 1. The proof follows in a similar manner described in the proof sketch in the main body of the paper. We start with proving Lemma 1 which establishes the convergence of the outer loop iterates in terms of inner loop convergence error. Subsequently, we establish the convergence of the inner loop. Finally we combine them to prove Theorem 1.

### C.1   OUTER LOOP CONVERGENCE ANALYSIS: PROOF OF LEMMA 1

In this subsection, we prove Lemma 1. The proof is inspired by the analysis in Chen et al. (2023) for non-robust Q-learning. The analysis of the outer loop follows from the paper (Chen et al., 2023). To write the bound for the outer loop, we have to start with a few notations as used in the mentioned paper. Recall, the function approximation error $\epsilon_{\text{approx}}$ is defined as:

$$\epsilon_{\text{approx}} := \sup_{Q=\text{Clip}(\Phi\theta), \theta \in \mathbb{R}^{n_\theta}} \left\| \text{Clip}\left(\Pi \mathcal{T}^{\text{rob},\pi}(Q)\right) - \mathcal{T}^{\text{rob},\pi}(Q) \right\|_\infty.$$

Also, recall the definition of $\theta_t^*$ from Equation 10.

Recall the fact that $Q^{\text{rob},\pi} = \mathcal{T}^{\text{rob},\pi}(Q^{\text{rob},\pi})$.

Then, for any $t = 1, 2, ..., T$, we have,

$$\|\hat{Q}_t - Q^{\mathrm{rob},\pi}\|_\infty = \|\mathrm{Clip}(\Phi\hat{\theta}_t) - \mathcal{T}^{\mathrm{rob},\pi}(Q^{\mathrm{rob},\pi})\|_\infty$$

$$= \underbrace{\|(\mathcal{T}^{\mathrm{rob},\pi}(\hat{Q}_{t-1}) - \mathcal{T}^{\mathrm{rob},\pi}(Q^{\mathrm{rob},\pi}))\|_\infty}_{I}$$

$$+ \underbrace{\|(\mathrm{Clip}(\Phi\hat{\theta}_t) - \mathrm{Clip}(\Pi\mathcal{T}^{\mathrm{rob},\pi}(\hat{Q}_{t-1})))\|_\infty}_{II}$$

$$+ \underbrace{\|(\mathcal{T}^{\mathrm{rob},\pi}(\hat{Q}_{t-1}) - \mathrm{Clip}(\Pi\mathcal{T}^{\mathrm{rob},\pi}(\hat{Q}_{t-1})))\|_\infty}_{\leq \epsilon_{\mathrm{approx}}}.$$

**First Term:**

$$I = \|(\mathcal{T}^{\mathrm{rob},\pi}(\hat{Q}_{t-1}) - \mathcal{T}^{\mathrm{rob},\pi}(Q^{\mathrm{rob},\pi}))\|_\infty \leq \gamma\|\hat{Q}_{t-1} - Q^{\mathrm{rob},\pi}\|_\infty,$$

as the robust bellman operator is a $\gamma$-contraction with respect to the $\infty$-norm (Iyengar, 2005).

**Second Term:**

$$II = \|(\mathrm{Clip}(\Phi\hat{\theta}_t) - \mathrm{Clip}(\Pi\mathcal{T}^{\mathrm{rob},\pi}(\hat{Q}_{t-1})))\|_\infty$$

$$\underbrace{\leq}_{(a)} \|\Phi\hat{\theta}_t - \Pi\mathcal{T}^{\mathrm{rob},\pi}(\hat{Q}_{t-1})\|_\infty$$

$$\underbrace{=}_{(b)} \||\Phi(\theta_{t-1,K} - \theta_{t-1}^*)\|_\infty$$

$$\leq \max_{s,a}\|\phi(s,a)\|_2\|\theta_{t-1,K} - \theta_{t-1}^*\|_2$$

$$\underbrace{\leq}_{(c)} \|\theta_{t-1,K} - \theta_{t-1}^*\|_2.$$

where $(a)$ is using the non-expansive property of the clipping operator with respect to $\|\cdot\|_\infty$; for $(b)$, recall the definition of $\theta_t^*$ in the inner loop in Equation 10; for $(c)$, assume $\|\phi(s,a)\|_2 \leq 1, \forall (s,a) \in \mathcal{S} \times \mathcal{A}$.

Hence, we get:

$$\|\hat{Q}_t - Q^{\mathrm{rob},\pi}\|_\infty \leq \gamma\|\hat{Q}_{t-1} - Q^{\mathrm{rob},\pi}\|_\infty + \|\theta_{t-1,K} - \theta_{t-1}^*\|_2 + \epsilon_{\mathrm{approx}}.$$

Unroll the recursion and take the expectation:

$$\mathbb{E}\|\hat{Q}_T - Q^{\mathrm{rob},\pi}\|_\infty \leq \gamma^T\|\hat{Q}_0 - Q^{\mathrm{rob},\pi}\|_\infty + \sum_{t=1}^{T}\gamma^{T-t}\mathbb{E}[\|\theta_{t-1,K} - \theta_{t-1}^*\|_2] + \frac{\epsilon_{\mathrm{approx}}}{1-\gamma}.$$

## C.2 INNER LOOP CONVERGENCE ANALYSIS

The purpose of this subsection is to bound the term $\mathbb{E}[\|\theta_{t-1,K} - \theta_{t-1}^*\|_2]$ for any fixed outer loop iteration index $t$.

Using the notations stated in the Lemma 2, we instantiate different problem-dependent constants, namely $C_G$ and $C_{F,M}$ as follows for differet uncertainty sets. In the fast time-scale, we have $\|\nu_{t,k}\|_2 \leq B_\nu$. This implies, $|\psi(s,a)^\top\nu_{t,k}| \leq B_\nu$ for all $(s,a)$ pair. Hence, for the rest of the Appendix denote $M := B_\nu$ and similarly $C_{F,M} := C_{B_\nu}$.

Recall from the Appendix B, we know that,

1. For total variation uncertainty sets, $C_{B_\nu} = (1+\delta)\left(M + \frac{1}{1-\gamma}\right)$.

2. For Wasserstein-$\ell$ uncertainty sets, $C_{B_\nu} = (1 + \delta^\ell)B_\nu + \frac{1}{1-\gamma}$.

Also, from the Appendix B, we know that,

1. For total variation uncertainty sets, $C_G = \max(\delta, 1 - \delta)$.
2. For Wasserstein-$\ell$ uncertainty sets, $C_G = 1 + \delta^l$.

In this subsection, we show that for each outer iteration $t$, the inner loop parameter $\theta_{t,k}$ converges to $\theta_t^*$ as defined in Equation (10). Recall the definition $F_{s,a}^{*,V} := \sup_{\lambda_s^a} F(\lambda_s^a; V, P_0(\cdot|s,a))$. We denote the inner loop error for outer iteration index $t$ as $e_{t,k} := \theta_{t,k} - \theta_t^*$ with

$$\theta_t^* := (\Phi^\top D^\pi \Phi)^{-1} \Phi^\top D^\pi \big[ r + \gamma F^{*, V_{\hat{\theta}_t}^{\mathrm{rob}}} \big]. \tag{22}$$

Using earlier notation, the dual objective corresponding to an $(s,a)$-pair for a target value function $V_{\hat{\theta}_t}^{\mathrm{rob}}$ is

$$\max_{\lambda_s^a} F(\lambda_s^a; V_{\hat{\theta}_t}^{\mathrm{rob}}, P_0(\cdot|s,a)).$$

For the rest of the discussion in this subsection, let us fix an outer loop iteration $t$ and the target parameter $\hat{\theta}_t$ is treated as a deterministic vector. For a given outer loop index $t$, for all inner loop iterations $k \geq 1$ let the filtration $\mathcal{F}_{t,k}$ be the sigma algebra generated by the transitions sampled till inner loop iteration index $k-1$ that is, on the set $\{S_{t,j}, A_{t,j}, S_{t,j+1} : 0 \leq j \leq k-1\}$.

Observe that the pair process $Z_{t,k} := (S_{t,k}, A_{t,k})$ is a Markov chain. We define another filtration $\mathcal{G}_{t,k}$ as the sigma algebra over the set $\{Z_{t,0}, Z_{t,1}, ..., Z_{t,k}\}$.

### C.2.1 ANALYSIS ON THE FAST TIME-SCALE:

The fast time-scale update is given as

$$\nu_{t,k+1} = \mathrm{Proj}_{\mathcal{M}_\nu}(\nu_{t,k} + \beta_k[\hat{G}(\psi(S_{t,k}, A_{t,k})^\top \nu_{t,k}; V_{\hat{\theta}_t}^{\mathrm{rob}}, S_{t,k+1})\psi(S_{t,k}, A_{t,k})]).$$

Define the diagonal matrix $D_{t,k} \in \mathbb{R}^{|\mathcal{S}||\mathcal{A}| \times |\mathcal{S}||\mathcal{A}|}$ with each diagonal element as $D_{t,k}((s,a),(s,a)) = \mathbf{1}_{(s,a)=(S_{t,k}, A_{t,k})}$.

For each outer iteration $t$ and dual vector $\nu_{t,k}$, we define a vector $\bar{g}_t(\nu_{t,k}) \in \mathbb{R}^{|\mathcal{S}||\mathcal{A}|}$, indexed by $(s,a) \in \mathcal{S} \times \mathcal{A}$, via

$$[\bar{g}_t(\nu_{t,k})]_{s,a} := \mathbb{E}_{S' \sim P_0(\cdot|s,a)}\big[\hat{G}\big(\psi(s,a)^\top \nu_{t,k}; V_{\hat{\theta}_t}^{\mathrm{rob}}, S'\big)\big].$$

Here $S'$ denotes the next-state random variable with distribution $P_0(\cdot \mid s,a)$.

Also, define the stochastic update vector $X_{t,k} \in \mathbb{R}^{|\mathcal{S}||\mathcal{A}|}$ defined as

$$[X_{t,k}]_{s,a} := \mathbf{1}_{(s,a)=(S_{t,k}, A_{t,k})} \cdot \hat{G}(\psi(s,a)^\top \nu_{t,k}; V_{\hat{\theta}_t}^{\mathrm{rob}}, S_{t,k+1})\psi(s,a).$$

We split the update into stationary drift and different noise terms as:

$$\begin{aligned} &\nu_{t,k+1} \\ =&\mathrm{Proj}_{\mathcal{M}_\nu}\left(\nu_{t,k} + \beta_k\left[\Psi^\top D^\pi \bar{g}_t(\nu_{t,k}) + \underbrace{X_{t,k} - \mathbb{E}[X_{t,k}|\mathcal{G}_{t,k}]}_{m_{k+1}^\nu} + \underbrace{\mathbb{E}[X_{t,k}|\mathcal{G}_{t,k}] - \Psi^\top D^\pi \bar{g}_t(\nu_{t,k})}_{\zeta_{k+1}^\nu}\right]\right). \end{aligned}$$

We see that,

$$\zeta_{k+1}^\nu = \mathbb{E}[X_{t,k}|\mathcal{G}_{t,k}] - \Psi^\top D^\pi \bar{g}_t(\nu_{t,k}).$$

So the update now becomes,

$$\nu_{t,k+1} = \mathrm{Proj}_{\mathcal{M}_\nu}\left(\nu_{t,k} + \beta_k\left[\Psi^\top D^\pi \bar{g}_t(\nu_{t,k}) + m_{k+1}^\nu + \zeta_{k+1}^\nu\right]\right).$$

In the above equation, $m_{k+1}^\nu$ denotes the state-innovation noise that is a martingale difference on the filtration $\mathcal{G}_{t,k}$.

Hence,

$$\mathbb{E}[m_{k+1}^\nu | \mathcal{G}_{t,k}] = 0.$$

We analyze the finite time convergence of the fast time-scale first. We show that the fast time-scale update is equivalent to performing a stochastic gradient super-gradient ascent on the following objective function:

$$L_t(\nu) := \sum_{s,a} d^\pi(s,a) F(\psi(s,a)^\top \nu; V_{\hat\theta_t}, P_0(\cdot|s,a)).$$

It is easy to show that $L_t(\nu)$ is concave on $\nu$. Let $\nu_t^*$ be one maximizer of the above objective function $L_t(\nu)$ in the domain $\mathcal{M}_\nu$. Notice that the fast time-scale update depends on the target parameter vector $\hat\theta_t$ and independent of the slow time-scale parameters for a fixed outer loop.

Our goal is to bound the sub-optimality gap of the dual objective (the weighted objective $L_t(\nu)$) for each iteration in the inner loop. We will be able to use the error in estimating the dual objective from the fast time-scale as a bias in the slow time-scale to get a sample complexity bound for the inner loop of the algorithm 1.

Let us define the dual objective sub-optimality at $\nu = \nu_{t,k}$ for the fast time-scale as :

$$L_{t,k} := \sum_{s,a} d^\pi(s,a)[F(\psi(s,a)^\top \nu_t^*; V_{\hat\theta_t}^{\mathrm{rob}}, P_0(\cdot|s,a)) - F(\psi(s,a)^\top \nu_{t,k}; V_{\hat\theta_t}^{\mathrm{rob}}, P_0(\cdot|s,a))].$$

Define the Lyapunov function for the fast time-scale as

$$e_{t,k}^\nu = \|\nu_{t,k} - \nu_t^*\|_2^2.$$

Using the non-expansiveness of projection,

$$e_{t,k+1}^\nu \le \|\nu_{t,k} + \beta_k \Psi^\top D^\pi \bar g_t(\nu_{t,k}) + \beta_k(m_{k+1}^\nu + \zeta_{k+1}^\nu) - \nu_t^*\|_2^2.$$

We can write:

$$\begin{aligned}
e_{t,k+1}^\nu \le{}& \|(\nu_{t,k} - \nu_t^*)\|_2^2 + 2\beta_k(\nu_{t,k} - \nu_t^*)^\top \Psi^\top D^\pi \bar g_t(\nu_{t,k}) \\
& + 2\beta_k(\nu_{t,k} - \nu_t^*)^\top(m_{k+1}^\nu + \zeta_{k+1}^\nu) \\
& + \beta_k^2 \|\Psi^\top D^\pi \bar g_t(\nu_{t,k}) + m_{k+1}^\nu + \zeta_{k+1}^\nu\|_2^2.
\end{aligned}$$

We simplify the term $(\nu_{t,k} - \nu_t^*)^\top \Psi^\top D^\pi \bar g_t(\nu_{t,k})$ first.

$$\begin{aligned}
(\nu_{t,k} - \nu_t^*)^\top \Psi^\top D^\pi \bar g_t(\nu_{t,k}) &= \sum_{s,a \in \mathcal{S} \times \mathcal{A}} d^\pi(s,a) \bar g_t(\nu_{t,k})(s,a)(\nu_{t,k} - \nu_t^*)^\top \psi(s,a) \\
&= (\nu_{t,k} - \nu_t^*)^\top \nabla_{\nu = \nu_{t,k}} L_t(\nu).
\end{aligned}$$

Hence, from the first order optimality condition on a concave objective, we can write:

$$(\nu_{t,k} - \nu_t^*)^\top \Psi^\top (D^\pi) \bar g_t(\nu_{t,k}) \le -L_{t,k}$$

Hence, we get,

$$\begin{aligned}
e_{t,k+1}^\nu \le{}& \|(\nu_{t,k} - \nu_t^*)\|_2^2 - 2\beta_k L_{t,k} \\
& + 2\beta_k(\nu_{t,k} - \nu_t^*)^\top(m_{k+1}^\nu) + 2\beta_k(\nu_{t,k} - \nu_t^*)^\top(\zeta_{k+1}^\nu) \\
& + \beta_k^2 \|\Psi^\top D^\pi \bar g_t(\nu_{t,k}) + m_{k+1}^\nu + \zeta_{k+1}^\nu\|_2^2.
\end{aligned}$$

Now we condition on a lagged filtration $\mathcal{G}_{t,k-\tau}$ where $\tau \leq k$ would be chosen later.

$$\mathbb{E}\left[e_{t,k+1}^{\nu}|\mathcal{G}_{t,k-\tau}\right] \leq \mathbb{E}\left[e_{t,k}^{\nu}|\mathcal{G}_{t,k-\tau}\right] - 2\beta_k\mathbb{E}\left[L_{t,k}|\mathcal{G}_{t,k-\tau}\right]$$
$$+ 2\beta_k\mathbb{E}\left[(\nu_{t,k} - \nu_t^*)^{\top}(m_{k+1}^{\nu})|\mathcal{G}_{t,k-\tau}\right] + 2\beta_k\mathbb{E}\left[(\nu_{t,k} - \nu_t^*)^{\top}(\zeta_{k+1}^{\nu})|\mathcal{G}_{t,k-\tau}\right]$$
$$+ \beta_k^2\mathbb{E}\left[\|\Psi^{\top}D^{\pi}\bar{g}_t(\nu_{t,k}) + m_{k+1}^{\nu} + \zeta_{k+1}^{\nu}\|_2^2|\mathcal{G}_{t,k-\tau}\right].$$

Let us first bound the $\beta_k^2$ terms. Recall from the conditions described in Lemma 2, $\|\bar{g}_t(\nu_{t,k})\|_{\infty} \leq C_G$. As $\|\psi(s,a)\|_2 \leq 1, \forall(s,a) \in \mathcal{S} \times \mathcal{A}$, one can easily show that $\|\Psi^{\top}D^{\pi}\bar{g}_t(\nu_{t,k})\|_2 \leq C_G$ and $\|m_{k+1}^{\nu}\|_2 \leq 2C_G$ and $\|\zeta_{k+1}^{\nu}\|_2 \leq 2C_G$. Hence,

$$\beta_k^2\mathbb{E}\left[\|\Psi^{\top}D^{\pi}\bar{g}_t(\nu_{t,k}) + m_{k+1}^{\nu} + \zeta_{k+1}^{\nu}\|_2^2|\mathcal{G}_{t,k-\tau}\right] \leq 25\beta_k^2 C_G^2.$$

Now we work on the cross terms. Let us start with $2\beta_k\mathbb{E}\left[(\nu_{t,k} - \nu_t^*)^{\top}D^{\pi}(m_{k+1}^{\nu})|\mathcal{G}_{t,k-\tau}\right].$

We write:

$$2\beta_k\mathbb{E}\left[(\nu_{t,k} - \nu_t^*)^{\top}D^{\pi}(m_{k+1}^{\nu})|\mathcal{G}_{t,k-\tau}\right] = 2\beta_k\mathbb{E}\left[\mathbb{E}\left[(\nu_{t,k} - \nu_t^*)^{\top}D^{\pi}(m_{k+1}^{\nu})|\mathcal{G}_{t,k}\right]|\mathcal{G}_{t,k-\tau}\right]$$
$$= 2\beta_k\mathbb{E}\left[(\nu_{t,k} - \nu_t^*)^{\top}D^{\pi}[\mathbb{E}\left[(m_{k+1}^{\nu})|\mathcal{G}_{t,k}\right]|\mathcal{G}_{t,k-\tau}\right]$$
$$= 0.$$

Now we focus on the term $2\beta_k\mathbb{E}\left[(\nu_{t,k} - \nu_t^*)^{\top}(\zeta_{k+1}^{\nu})|\mathcal{G}_{t,k-\tau}\right].$

We use the shorthand
$$Z_{t,k} := (S_{t,k}, A_{t,k}) \in \mathcal{S} \times \mathcal{A}$$
for the state-action pair at outer iteration $t$ and inner iteration $k$.

Define the vector $e_{Z_{t,k}}$

$$e_{Z_{t,k}}(s,a) := \mathbf{1}_{S_{t,k}, A_{t,k}=s,a}.$$

Recall the definition

$$\eta_k^{t,\tau}(\cdot) := \mathbb{P}\left((S_{t,k}, A_{t,k}) \in \cdot \mid S_{t,0}, A_{t,0}, \ldots, S_{t,k-\tau}, A_{t,k-\tau}\right)$$

From Assumption 1, we know the following holds:
$$\|\eta_k^{t,\tau} - d^{\pi}\|_{\text{TV}} \leq C_{\text{mix}}\rho^{\tau} \qquad (0 < \rho < 1).$$

Thus, we can write:

$$2\beta_k\mathbb{E}\left[(\nu_{t,k} - \nu_t^*)^{\top}(\Psi^{\top}(e_{Z_{t,k}} - d^{\pi}) \odot \bar{g}_t(\nu_{t,k}))|\mathcal{G}_{t,k-\tau}\right]$$
$$= 2\beta_k\mathbb{E}\left[(\Psi(\nu_{t,k} - \nu_t^*))^{\top}((e_{Z_{t,k}} - d^{\pi}) \odot \bar{g}_t(\nu_{t,k}))|\mathcal{G}_{t,k-\tau}\right]$$
$$\leq 2\beta_k\mathbb{E}\left[\|(\Psi(\nu_{t,k} - \nu_t^*))\|_{\infty}\|((e_{Z_{t,k}} - d^{\pi}) \odot \bar{g}_t(\nu_{t,k}))\|_1|\mathcal{G}_{t,k-\tau}\right]$$
$$\leq 2\beta_k\mathbb{E}\left[2B_{\nu}C_G\|(e_{Z_{t,k}} - d^{\pi})\|_1|\mathcal{G}_{t,k-\tau}\right]$$
$$\leq 8\beta_k B_{\nu}C_G C_{\text{mix}}\rho^{\tau}.$$

Putting it together, we have:

$$\mathbb{E}\left[e_{t,k+1}^{\nu}|\mathcal{G}_{t,k-\tau}\right] \leq \mathbb{E}\left[e_{t,k}^{\nu}|\mathcal{G}_{t,k-\tau}\right] - 2\beta_k\mathbb{E}\left[L_{t,k}|\mathcal{G}_{t,k-\tau}\right] + 8\beta_k B_{\nu}C_G C_{\text{mix}}\rho^{\tau} + 25\beta_k^2 g_m^2.$$

Now we choose $\forall l \in (\lfloor k/2 \rfloor, k-1)$, $\tau = \tau_{\beta_l} := \lceil \frac{log(\frac{C_{\text{mix}}}{\beta_0}\sqrt{l+1})}{log(\frac{1}{\rho})} \rceil$, and assume $k \geq 2\tau_{\beta_k}$ then, taking unconditional expectation gives us: (as $C_{\text{mix}}\rho^{\tau_{\beta_l}} \leq \beta_l$) $\forall l \in (\lfloor k/2 \rfloor, k-1)$:
$$2\beta_k\mathbb{E}[L_l] \leq \mathbb{E}[e_{t,l}^{\nu}] - \mathbb{E}[e_{t,l+1}^{\nu}] + \beta_l^2(8B_{\nu}C_G + 25C_G^2).$$

Next, we use telescoping for iterates over the index $l$ from $\lfloor \frac{k}{2} \rfloor$ to $k-1$.

$$2 \sum_{l=\lfloor k/2 \rfloor}^{k-1} \beta_l \mathbb{E}[L_l] \leq 4B_\nu^2 + (8B_\nu C_G + 25C_G^2) \sum_{l=\lfloor k/2 \rfloor}^{k-1} \beta_l^2, \tag{23}$$

where, we used that $\|e_{t,\lfloor k/2 \rfloor}\|_2 \leq 4B_\nu^2$ due to the projection step $\mathrm{Proj}_{\mathcal{M}_\nu}$. Recall, the fast time-scale passes the following suffix-average of the dual parameter vector iterates to the slow time-scale at each iterate $k$:

$$\bar{\nu}_{t,k} = \frac{1}{\lceil k/2 \rceil} \sum_{\lfloor k/2 \rfloor}^{k-1} \nu_{t,k}.$$

We use the step-size rule of $\beta_k = \frac{\beta_0}{\sqrt{k+1}}$.

Similar to the definition of $L_{t,k}$, let us define

$$\overline{L}_{t,k} := \sum_{s,a} d^\pi(s,a)[F(\psi(s,a)^\top \nu_t^*; V_{\hat{\theta}_t}^{\mathrm{rob}}, P_0(\cdot|s,a)) - F(\psi(s,a)^\top \bar{\nu}_{t,k}; V_{\hat{\theta}_t}^{\mathrm{rob}}, P_0(\cdot|s,a))] \tag{24}$$

Hence, using Jensen's inequality, we write:

$$\mathbb{E}[\overline{L}_{t,k}] \leq \frac{1}{\lceil k/2 \rceil} \sum_{l=\lfloor k/2 \rfloor}^{k-1} \mathbb{E}[L_{t,l}]$$

$$\underbrace{\leq}_{(a)} \frac{1}{\lceil k/2 \rceil} \frac{\sqrt{k}}{\beta_0} \sum_{l=\lfloor k/2 \rfloor}^{k-1} \beta_l \mathbb{E}[L_{t,k}] \leq \frac{2}{k} \frac{\sqrt{k}}{\beta_0} \sum_{l=\lfloor k/2 \rfloor}^{k-1} \beta_l \mathbb{E}[L_{t,l}]$$

$$\underbrace{\leq}_{(b)} \frac{4B_\nu^2}{\beta_0 \sqrt{k}} + \frac{(8B_\nu C_G + 25C_G^2)}{\beta_0 \sqrt{k}} \sum_{l=\lfloor k/2 \rfloor}^{k-1} \beta_l^2$$

$$= \frac{4B_\nu^2}{\beta_0 \sqrt{k}} + \frac{(8B_\nu C_G + 25C_G^2)}{\beta_0 \sqrt{k}} \sum_{l=\lfloor k/2 \rfloor}^{k-1} \frac{\beta_0^2}{l+1}$$

$$\underbrace{\leq}_{(c)} \frac{4B_\nu^2}{\beta_0 \sqrt{k}} + \frac{(8B_\nu C_G + 25C_G^2)\left(\beta_0^2(1 + \ln(k) - \ln(k/2))\right)}{\beta_0 \sqrt{k}}$$

$$\leq \frac{C_{fast}}{\sqrt{k}}$$

where

$$C_{fast} = \frac{(4B_\nu^2 + \beta_0^2(8B_\nu C_G + 25C_G^2)\ln(2e))}{\beta_0}. \tag{25}$$

In $(a)$, we used $\beta_k \geq \frac{\beta_0}{\sqrt{k}}, \forall k \leq k-1$. In $(b)$, we used Equation 23. In $(c)$, we used the following identity: $\ln(k) \leq 1 + \frac{1}{2} + ... + \frac{1}{k} \leq 1 + \ln(k)$ .

In summary, we have the following guarantee from the fast time-scale:

**Lemma 4.** *Fix an outer loop $t \geq 0$. The following holds for the fast time-scale iterates of the Algorithm 1: If $k \geq 2 \left\lceil \frac{log(\frac{C_{\mathrm{mix}}}{\beta_0}\sqrt{k+1})}{log(\frac{1}{\rho})} \right\rceil$,*

$$\mathbb{E}[\overline{L}_{t,k}] \leq \frac{C_{fast}}{\sqrt{k}}, \tag{26}$$

*where, $\overline{L}_{t,k}$ is defined in Equation (24) and $C_{fast}$ is given in Equation (25).*

### C.2.2 Slow time-scale Analysis

Next, we will prove the convergence of the slow time-scale iterates $\theta_{t,k}$ to $\theta_t^*$ for sufficiently large $k$. We denote the inner loop error for outer iteration index $t$ as $e_{t,k} := \theta_{t,k} - \theta_t^*$ with

$$\theta_t^* := (\Phi^\top D^\pi \Phi)^{-1} \Phi^\top D^\pi \big[ r + \gamma \, F^{*, V_{\hat\theta_t}^{\mathrm{rob}}} \big] \tag{27}$$

Recall the notation $Z_{t,k} = (S_{t,k}, A_{t,k})$. The slow update is

$$\theta_{t,k+1} = \theta_{t,k} + \alpha_k \, TD_{t,k+1} \, \phi(Z_{t,k}),$$

$$TD_{t,k+1} = R(Z_{t,k}) + \gamma \, \hat{F}(\psi(Z_{t,k})^\top \bar\nu_{t,k}; V_{\hat\theta}^{\mathrm{rob}}, S_{t,k+1}) - \phi(Z_{t,k})^\top \theta_{t,k},$$

with $\bar\nu_{t,k}$ the suffix average produced by the fast time-scale updates.

For each outer iteration $t$ and dual vector $\nu_{t,k}$, we define a vector $\bar{f}_t(\nu_{t,k}) \in \mathbb{R}^{|\mathcal{S}||\mathcal{A}|}$, indexed by $(s,a) \in \mathcal{S} \times \mathcal{A}$, via

$$[\bar{f}_t(\nu_{t,k})]_{s,a} := \mathbb{E}_{S' \sim P_0(\cdot | s,a)} \big[ \hat{F}\big( \psi(s,a)^\top \nu_{t,k}; V_{\hat\theta_t}^{\mathrm{rob}}, S' \big) \big].$$

Here $S'$ denotes the next-state random variable with distribution $P_0(\cdot \mid s, a)$.

We decompose the term $TD_{t,k+1}\phi(Z_{t,k})$ as

$$H_{t,k}^\theta \;+\; b_{t,k}^\theta \;+\; \xi_{t,k+1}^\theta \;+\; m_{t,k+1}^\theta,$$

where

$$H_{t,k}^\theta := \Phi^\top D^\pi \big[ r + \gamma F^{*, V_{\hat\theta_t}^{\mathrm{rob}}} - \Phi\theta_{t,k} \big] \underbrace{=}_{\text{from Equation (10)}} \Phi^\top D^\pi \Phi(\theta_t^* - \theta_{t,k}),$$

$$b_{t,k}^\theta := \gamma \Phi^\top D^\pi \Big[ F^{V_{\hat\theta_t}^{\mathrm{rob}}}(\bar\nu_{t,k}) - F^{*, V_{\hat\theta_t}^{\mathrm{rob}}} \Big],$$

$$\xi_{t,k+1}^\theta := \Phi^\top \big( D_{t,k} - D^\pi \big) \Big[ r + \gamma F^{V_{\hat\theta_t}^{\mathrm{rob}}}(\bar\nu_{t,k}) - \Phi\theta_{t,k} \Big]$$

$$= \Phi^\top \big( D_{t,k} - D^\pi \big) \Big[ r + \gamma \bar{f}_t(\bar\nu_{t,k}) - \Phi\theta_{t,k} \Big],$$

$$m_{t,k+1}^\theta := \gamma \, \Phi^\top D_{t,k} \Big( \hat{F}(\psi(Z_{t,k})^\top \bar\nu_{t,k}; S_{k+1}, V_{\hat\theta_t}^{\mathrm{rob}}) - \bar{f}_t(\bar\nu_{t,k}) \Big).$$

Note $\mathbb{E}[m_{t,k+1}^\theta \mid \mathcal{G}_{t,k}] = 0$ and, by tower property of conditional expectation, $\mathbb{E}[e_{t,k}^\top m_{t,k+1}^\theta \mid \mathcal{G}_{t,k-\tau}] = 0$ for a lag $\tau \le k$.

Recall the definition

$$\eta_k^{t,\tau}(\cdot) := \mathbb{P}\big( (S_{t,k}, A_{t,k}) \in \cdot \,\big|\, S_{t,0}, A_{t,0}, \dots, S_{t,k-\tau}, A_{t,k-\tau} \big)$$

From Assumption 1, we know the following holds:

$$\|\eta_k^{t,\tau} - d^\pi\|_{\mathrm{TV}} \le C_{\mathrm{mix}} \, \rho^\tau \qquad (0 < \rho < 1).$$

For the fixed lag (to be chosen later) $\tau \ge 1$ and define $\mathcal{H}_{t,k}$ as the sigma algebra over the set $\{\mathcal{G}_{t,k-\tau}, \theta_{t,k}, \bar\nu_{t,k}\}$.

Conditioning on $\mathcal{H}_{t,k}$ "freezes" $e_{t,k} := \theta_{t,k} - \theta_t^*$ and

$$y_{t,k} := r + \gamma \, \bar{f}_t(\bar\nu_{t,k}) - \Phi\theta_{t,k}.$$

We use: for any signed vector $w$ on $\mathcal{S} \times \mathcal{A}$,

$$\big\|\Phi^\top w\big\|_2 = \Big\| \sum_z w_{s,a}\, \phi(s,a) \Big\|_2 \le \sum_{s,a} |w_{s,a}| = \|w\|_1, \tag{28}$$

because each row vector satisfies $\|\phi(s,a)\|_2 \le 1$. We also write the Markov noise term in terms of $y_{t,k}$ as

$$\xi_{t,k+1}^\theta \;=\; \Phi^\top \big(D_{t,k} - D^\pi\big)\, y_{t,k}.$$

Finally set $Y_0 := 1 + \gamma C_{F,M} + \frac{1}{(1-\gamma)\sqrt{\mu}}$ so that

$$\|y_{t,k}\|_\infty \;\le\; Y_0 + \|e_{t,k}\|_2, \tag{29}$$

using Lemma 7 and, $r(\cdot) \in [0,1]$, $\|\bar{f}_t(\cdot)\|_\infty \le C_{F,M}$, and $\|\phi(z)\|_2 \le 1$.

Recall the definition $e_{t,k} = \theta_{t,k} - \theta_t^*$ with

$$\theta_t^* := (\Phi^\top D^\pi \Phi)^{-1}\Phi^\top D^\pi \big[r + \gamma\, F^{*,V_{\hat{\theta}_t}^{\mathrm{rob}}}\big].$$

(I) ONE-STEP LYAPUNOV EXPANSION UNDER A CONDITIONAL EXPECTATION WITH A FILTRATION UNDER A GENERIC LAG $\tau$

With $x_{t,k} := \|e_{t,k}\|^2$, for $k \ge \tau$, as $\mathbb{E}[e_{t,k}^\top m_{t,k+1}^\theta \mid \mathcal{G}_{t,k-\tau}] = 0$, we can write, if $k \ge \tau$

$$
\begin{aligned}
\mathbb{E}[x_{k+1} \mid \mathcal{G}_{t,k-\tau}] &= \mathbb{E}\Big[\big\|e_{t,k} + \alpha_k(H_{t,k}^\theta + b_{t,k}^\theta + \xi_{t,k+1}^\theta + m_{t,k+1}^\theta)\big\|^2 \,\Big|\, \mathcal{G}_{t,k-\tau}\Big] \\
&= x_{t,k} + 2\alpha_k\,\mathbb{E}\big[e_{t,k}^\top H_{t,k}^\theta \,\big|\, \mathcal{G}_{t,k-\tau}\big] + 2\alpha_k\,\mathbb{E}\big[e_{t,k}^\top b_{t,k}^\theta \,\big|\, \mathcal{G}_{t,k-\tau}\big] \\
&\quad + 2\alpha_k\,\mathbb{E}\big[e_{t,k}^\top \xi_{t,k+1}^\theta \,\big|\, \mathcal{G}_{t,k-\tau}\big] \\
&\quad + \alpha_k^2\,\mathbb{E}\big[\|H_{t,k}^\theta + b_{t,k}^\theta + \xi_{t,k+1}^\theta + m_{t,k+1}^\theta\|^2 \,\big|\, \mathcal{G}_{t,k-\tau}\big].
\end{aligned}
\tag{30}
$$

(II) MAIN DRIFT

Recall, we denote $\mu$ as the minimum eigenvalue of the matrix $\Phi^\top D^\pi \Phi$ and from Assumption 1, $\mu > 0$.

Since $e_{t,k}^\top H_{t,k}^\theta = -e_{t,k}^\top(\Phi^\top D^\pi \Phi)e_{t,k} \le -\mu\,\|e_{t,k}\|^2$,

$$2\alpha_k\,\mathbb{E}\big[e_{t,k}^\top H_{t,k}^\theta \,\big|\, \mathcal{G}_{t,k-\tau}\big] \;\le\; -2\mu\,\alpha_k\,\mathbb{E}[x_{t,k} \mid \mathcal{G}_{t,k-\tau}]. \tag{31}$$

(III) CROSS TERM CORRESPONDING TO BIAS

By conditional Cauchy–Schwarz and Young inequality,

$$2\alpha_k\,\mathbb{E}\big[e_{t,k}^\top b_{t,k}^\theta \,\big|\, \mathcal{G}_{t,k-\tau}\big] \;\le\; \frac{\mu}{2}\,\alpha_k\,\mathbb{E}[x_{t,k} \mid \mathcal{G}_{t,k-\tau}] \;+\; \frac{2\alpha_k}{\mu}\,\mathbb{E}\big[\|b_{t,k}^\theta\|^2 \,\big|\, \mathcal{G}_{t,k-\tau}\big]. \tag{32}$$

(IV) CROSS TERM CORRESPONDING TO MARKOV NOISE

**Lemma 5** (Cross with Markov noise). *For any $\tau \ge 1$, if $k \ge \tau$*

$$
\begin{aligned}
2\alpha_k\,\mathbb{E}\big[e_{t,k}^\top \xi_{t,k+1}^\theta \,\big|\, \mathcal{G}_{t,k-\tau}\big] &\le \Big(\frac{\mu}{2} + 4\min(1, C_{\mathrm{mix}}\rho^\tau)\Big)\,\alpha_k\,\mathbb{E}\big[\|e_{t,k}\|_2^2 \,\big|\, \mathcal{G}_{t,k-\tau}\big] \\
&\quad + \frac{8\,Y_0^2}{\mu}\,\alpha_k\,\min(1, C_{\mathrm{mix}}\rho^\tau)^2.
\end{aligned}
$$

*Proof.* By the tower property,

$$\mathbb{E}\big[e_{t,k}^\top \xi_{t,k+1}^\theta \,\big|\, \mathcal{G}_{t,k-\tau}\big] = \mathbb{E}\big[\,\mathbb{E}\big[e_{t,k}^\top \Phi^\top(D_{t,k} - D^\pi)\,y_{t,k} \,\big|\, \mathcal{H}_{t,k}\big] \,\big|\, \mathcal{G}_{t,k-\tau}\big].$$

Given $\mathcal{H}_{t,k}$, the only randomness is $Z_{t,k} \sim \eta_k^{t,\tau}$. Hence

$$\mathbb{E}\big[\Phi^\top(D_{t,k} - D^\pi)\,y_{t,k} \,\big|\, \mathcal{H}_{t,k}\big] = \Phi^\top(\eta_k^{t,\tau} - d^\pi) \odot y_{t,k}.$$

Therefore,

$$\Big|\mathbb{E}\big[e_{t,k}^\top \xi_{t,k+1}^\theta \,\big|\, \mathcal{G}_{t,k-\tau}\big]\Big| \le \mathbb{E}\big[\,\|e_{t,k}\|_2\,\big\|\Phi^\top(\eta_k^{t,\tau} - d^\pi) \odot y_{t,k}\big\|_2 \,\big|\, \mathcal{G}_{t,k-\tau}\big].$$

Apply Equation (28) and $\|(\eta_k^{t,\tau} - d^\pi) \odot y_{t,k}\|_1 \leq \|\eta_k^{t,\tau} - d^\pi\|_1 \|y_{t,k}\|_\infty \leq 2\min(1, C_{\mathrm{mix}}\rho^\tau)\|y_{t,k}\|_\infty$:

$$\left| \mathbb{E}[e_{t,k}^\top \xi_{t,k+1}^\theta \mid \mathcal{G}_{t,k-\tau}] \right| \leq 2\min(1, C_{\mathrm{mix}}\rho^\tau) \mathbb{E}[\|e_{t,k}\|_2 \|y_{t,k}\|_\infty \mid \mathcal{G}_{t,k-\tau}].$$

Multiply by $2\alpha_k$ and split $\|y_{t,k}\|_\infty$ using Equation (29):

$$2\alpha_k |\mathbb{E}[\cdot]| \leq 4\alpha_k \min(1, C_{\mathrm{mix}}\rho^\tau) \mathbb{E}[\|e_{t,k}\|_2 Y_0 \mid \mathcal{G}_{t,k-\tau}] + 4\alpha_k \min(1, C_{\mathrm{mix}}\rho^\tau) \mathbb{E}[\|e_{t,k}\|_2 \|e_{t,k}\|_2 \mid \mathcal{G}_{t,k-\tau}].$$

For the first term in the right hand side, use Young's inequality:

$$4\alpha_k \min(1, C_{\mathrm{mix}}\rho^\tau) Y_0 \|e_{t,k}\|_2 \leq \alpha_k \left( \frac{\mu}{2}\|e_{t,k}\|_2^2 + \frac{8\min(1, C_{\mathrm{mix}}\rho^\tau)^2 Y_0^2}{\mu} \right).$$

Combining,

$$2\alpha_k \mathbb{E}[e_{t,k}^\top \xi_{t,k+1}^\theta \mid \mathcal{G}_{t,k-\tau}] \leq \left( \frac{\mu}{2} + 4\min(1, C_{\mathrm{mix}}\rho^\tau) \right) \alpha_k \mathbb{E}[\|e_{t,k}\|_2^2 \mid \mathcal{G}_{t,k-\tau}] + \frac{8Y_0^2}{\mu}\alpha_k \min(1, C_{\mathrm{mix}}\rho^\tau)^2.$$

$\square$

(V) REMAINING SECOND ORDER TERMS

Now from the fact that $\|\phi(s,a)\|_2^2 \leq 1$, and from the conditions described in Lemma 2, we can write

$$\|H_{t,k}^\theta\|^2 \leq x_{t,k},$$

$$\mathbb{E}[\|m_{t,k+1}^\theta\|^2 \mid \mathcal{G}_{t,k-\tau}] \leq 4\gamma^2 C_{F,M}^2.$$

$$\mathbb{E}[\|\xi_{t,k+1}^\theta\|^2 \mid \mathcal{G}_{t,k-\tau}] \leq 8\mathbb{E}[x_{t,k} \mid \mathcal{G}_{t,k-\tau}] + 8Y_0^2$$

Therefore,

$$\alpha_k^2 \mathbb{E}[\|H_{t,k}^\theta + b_{t,k}^\theta + \xi_{t,k+1}^\theta + m_{t,k+1}^\theta\|^2 \mid \mathcal{G}_{t,k-\tau}]$$

$$\leq \alpha_k^2 \left( 68\mathbb{E}[x_{t,k} \mid \mathcal{G}_{t,k-\tau}] + 32\gamma^2 C_{F,M}^2 + 32Y_0^2 + 2\mathbb{E}[\|b_{t,k}^\theta\|^2 \mid \mathcal{G}_{t,k-\tau}] \right). \tag{33}$$

**Lemma 6** (Bias second order at $1/k$)**.** *For $k \geq \max(\tau, 2\tau_{\beta_k})$,*

$$\mathbb{E}[\|b_{t,k}^\theta\|_2^2 \mid \mathcal{G}_{t,k-\tau}] \leq \frac{2C_{\mathrm{bias}}}{k} + 2(\epsilon_{approx}^{dual})^2,$$

*with the explicit constant*

$$C_{\mathrm{bias}} := C_{\mathrm{fast}}^2 + 2C_{F,M}^2 + 64\,C_{F,M}^2 \frac{C_{\mathrm{mix}}\rho}{1-\rho},$$

*and $C_{\mathrm{fast}}$ as in Lemma 4.*

*Proof.* Using $\|\phi(s,a)\|_2 \leq 1$ for all $(s,a)$, we can write using Equation (28),

$$\|b_{t,k}^\theta\|_2 \leq \gamma \sum_{s,a} d^\pi(s,a) \left| F_{\hat\theta_t}^{V_{\hat\theta_t}^{\mathrm{rob}}}(\bar\nu_{t,k})_{s,a} - F_{s,a}^{*,V_{\hat\theta_t}^{\mathrm{rob}}} \right|$$

$$= \gamma \sum_{s,a} d^\pi(s,a) \left( F_{s,a}^{*,V_{\hat\theta_t}^{\mathrm{rob}}} - F_{\hat\theta_t}^{V_{\hat\theta_t}^{\mathrm{rob}}}(\bar\nu_{t,k})_{s,a} \right)$$

$$\leq \gamma \underbrace{\inf_{\nu \in \mathcal{M}_\nu} \sum_{s,a} d^\pi(s,a) \left( F_{s,a}^{*,V_{\hat\theta_t}^{\mathrm{rob}}} - F_{\hat\theta_t}^{V_{\hat\theta_t}^{\mathrm{rob}}}(\nu)_{s,a} \right)}_{\leq \epsilon_{\mathrm{approx}}^{\mathrm{dual}}} + \gamma \underbrace{\overline{L}_{t,k}}_{\text{fast-scale objective gap}}.$$

$$\mathbb{E}\big[\|b_{t,k}^\theta\|_2^2 \,\big|\, \mathcal{G}_{t,k-\tau}\big] \;\leq\; 2\mathbb{E}\big[(\overline{L}_{t,k})^2 \,\big|\, \mathcal{G}_{t,k-\tau}\big] + 2(\epsilon_{approx}^{dual})^2.$$

To bound the RHS at the $1/k$ scale, use the suffix average $\overline{L}_{t,k} \leq \frac{1}{m}\sum_{j=k-m}^{k-1} L_{t,j}$ with $m = \lfloor k/2\rfloor$. We also use from Section B that and $0 \leq L_{t,j} \leq 2C_{F,M}$. Write

$$\mathbb{E}\big[(\overline{L}_{t,k})^2 \,\big|\, \mathcal{G}_{t,k-\tau}\big] \leq \frac{1}{m^2}\bigg\{ \underbrace{\mathrm{Var}\bigg( \sum_{j=k-m}^{k-1} L_{t,j} \,\bigg|\, \mathcal{G}_{t,k-\tau} \bigg)}_{(I)} + \underbrace{\bigg( \mathbb{E}\sum_{j=k-m}^{k-1} L_{t,j}|\mathcal{G}_{t,k-\tau} \bigg)^2}_{(II)} \bigg\}.$$

*Term (II) (mean square).* By Lemma 4, for $k \geq \tau_{\beta_k}$, $\frac{1}{m}\mathbb{E}\sum_{j=k-m}^{k-1} L_{t,j} \leq \mathbb{E}[\overline{L}_{t,k}] \leq C_{\mathrm{fast}}/\sqrt{k}$, so $(II)/m^2 \leq C_{\mathrm{fast}}^2/k$.

*Term (I) (variance).* Under geometric mixing of the underlying markov chain and $0 \leq L_{t,j} \leq 2C_{F,M}$, the conditional covariances obey $|\mathrm{Cov}(L_{t,j}, L_{t,j+h} \,|\, \mathcal{G}_{t,k-\tau})| \leq 16C_{F,M}^2 C_{\mathrm{mix}}\rho^h$. Thus

$$\mathrm{Var}\bigg( \sum_{j=k-m}^{k-1} L_{t,j} \,\bigg|\, \mathcal{G}_{t,k-\tau} \bigg) = \sum_j \mathrm{Var}(L_{t,j} \,|\, \mathcal{G}_{t,k-\tau}) + 2\sum_{h=1}^{m-1}(m-h)\,\mathrm{Cov}(L_{t,j}, L_{t,j+h} \,|\, \mathcal{G}_{t,k-\tau})$$

$$\leq m\,C_{F,M}^2 + 32C_{F,M}^2\, m\sum_{h=1}^{\infty} C_{\mathrm{mix}}\rho^h \;\leq\; m\Big(C_{F,M}^2 + 32C_{F,M}^2\,\frac{C_{\mathrm{mix}}\rho}{1-\rho}\Big).$$

Dividing by $m^2$ and using $m \geq k/2$ gives

$$\frac{(I)}{m^2} \;\leq\; \frac{2}{k}\Big(C_{F,M}^2 + 32C_{F,M}^2\,\frac{C_{\mathrm{mix}}\rho}{1-\rho}\Big).$$

Combining the two terms yields

$$\mathbb{E}\big[(\overline{L}_{t,k})^2 \,\big|\, \mathcal{G}_{t,k-\tau}\big] \;\leq\; \frac{C_{\mathrm{fast}}^2}{k} + \frac{2}{k}\Big(C_{F,M}^2 + 32C_{F,M}^2\,\frac{C_{\mathrm{mix}}\rho}{1-\rho}\Big),$$

which is the claimed bound with the displayed $C_{\mathrm{bias}}$. $\square$

FINAL RECURSION FOR THE SLOW TIME-SCALE

From Equation (30) and the statements of the Lemma 5, Lemma 6 and from the analysis above, we can write the following recursion for $\|e_{t,k}\|_2$.

We first characterize a suitable choice of the lag $\tau$ to use in the recursion for $\|e_{t,k}\|_2$. We make a few observations. If $\tau \geq \big\lceil \frac{\log\big(C_{\mathrm{mix}}(k+1)/(c)\big)}{\log(1/\rho)} \big\rceil$ then $C_{\mathrm{mix}}\rho^\tau \leq \alpha_k$ .

Also, if $\tau \geq \tau_\mu := \big\lceil \frac{\log\big(C_{\mathrm{mix}}\frac{1}{\mu}\big)}{\log(1/\rho)} \big\rceil$, then $C_{\mathrm{mix}}\rho^\tau \leq \mu$.

From the above two observations, a suitable choice for $\tau$ in each inner iteration $k$ is $\tau_{\alpha_k} := \max\{\tau_{\alpha_k}, \tau_\mu\}$. As we are conditioning on filtration $\mathcal{G}_{t,k-\tau}$, we need $k \geq \tau_k$. Moreover, to use Lemma 4, we need $k \geq 2\tau_{\beta_k} = \big\lceil \frac{log(\frac{C_{\mathrm{mix}}}{\beta_0}\sqrt{k+1})}{log(\frac{1}{\rho})} \big\rceil$.

Hence, we use the following definitions:

$$\tau_{\alpha_k} := \Big\lceil \frac{\log\big(C_{\mathrm{mix}}(k+1)/c\big)}{\log(1/\rho)} \Big\rceil, \tag{34}$$

$$\tau_{\beta_k} := \Big\lceil \frac{\log\big(\frac{C_{\mathrm{mix}}}{\beta_0}\sqrt{k+1}\big)}{\log(1/\rho)} \Big\rceil, \tag{35}$$

$$\tau_\mu := \Big\lceil \frac{\log\big(C_{\mathrm{mix}}/\mu\big)}{\log(1/\rho)} \Big\rceil \tag{36}$$

$\forall k \geq \max(\tau_{\alpha_k}, 2\tau_{\beta_k}, \tau_\mu) :$

$$\mathbb{E}\big[\|e_{t,k+1}\|_2^2 \,\big|\, \mathcal{G}_{t,k-\tau}\big] \leq \left(1 - \mu\,\alpha_k + C_e\,\alpha_k^2\right) \mathbb{E}\big[\|e_{t,k}\|_2^2 \mathcal{G}_{t,k-\tau}\big]$$
$$+ \left(\tfrac{2\alpha_k}{\mu} + 2\alpha_k^2\right)\left(\frac{2C_{bias}}{k} + 2(\epsilon_{approx}^{dual})^2\right) + C_{cross}\alpha_k^2 + C_0\,\alpha_k^2,$$

and hence, after taking total expectation, $\forall k \geq \max(\tau, 2\tau_{\beta_k}, \tau_\mu)$

$$\mathbb{E}\|e_{t,k+1}\|_2^2 \leq \left(1 - \mu\,\alpha_k + C_e\,\alpha_k^2\right)\mathbb{E}\|e_{t,k}\|_2^2 +$$
$$+ \left(\tfrac{2\alpha_k}{\mu} + 2\alpha_k^2\right)\left(\frac{2C_{bias}}{k} + 2(\epsilon_{approx}^{dual})^2\right) + C_{cross}\,\alpha_k^2 + C_0\,\alpha_k^2. \tag{37}$$

The constants in the above recursion are given as:

$$C_e := 72,$$

$$C_0 := 32\gamma^2 C_{F,M}^2 + 32\left(1 + \gamma C_{F,M} + \frac{1}{(1-\gamma)\sqrt{\mu}}\right)^2,$$

$$C_{cross} := 8\left(1 + \gamma C_{F,M} + \frac{1}{(1-\gamma)\sqrt{\mu}}\right)^2,$$

$$C_{bias} := C_{fast}^2 + 2C_{F,M}^2 + 64\,C_{F,M}^2\,\frac{C_{mix}\rho}{1-\rho},$$

$$C_{fast} := \frac{(4B_\nu^2 + \beta_0^2(8B_\nu C_G + 25C_G^2))\ln(2e))}{\beta_0}.$$

Now we derive the last iterate convergence of the error $\|e_{t,k+1}^\theta\|_2$.

We also know from the definition of $\varepsilon_{approx}^{dual}$,

$$|\varepsilon_{approx}^{dual}| \leq \frac{1}{1-\gamma} + C_{F,M}.$$

In compact notations, $\forall k \geq \max(\tau_{\alpha_k}, 2\tau_{\beta_k}, \tau_\mu)$

$$\mathbb{E}\|e_{t,k+1}\|_2^2 \leq \left(1 - \mu\,\alpha_k + C_e\,\alpha_k^2\right)\mathbb{E}\|e_{t,k}\|_2^2 +$$
$$\alpha_k^2\left(\left(\tfrac{4}{c\mu} + 2\right)2C_{bias} + C_{cross} + C_0 + 4\left(\frac{1}{1-\gamma} + C_{F,M}\right)^2\right)$$
$$+ \frac{4\alpha_k(\epsilon_{approx}^{dual})^2}{\mu}$$

Or,

$$\mathbb{E}\|e_{t,k+1}\|_2^2 \leq \left(1 - \mu\,\alpha_k + C_e\,\alpha_k^2\right)\mathbb{E}\|e_{t,k}\|_2^2 + \alpha_k^2 C_1 + \frac{4\alpha_k(\epsilon_{approx}^{dual})^2}{\mu} \tag{38}$$

with $C_1 = \left(\left(\tfrac{8}{c\mu} + 4\right)C_{bias} + C_{cross} + C_0 + 4\left(\frac{1}{1-\gamma} + C_{F,M}\right)^2\right).$

Next, we derive the final iterate convergence from the above recursion for the step-size rule $\alpha_k = \frac{c}{1+k}, \quad \forall k \in [0, K-1]$:

## C.3 LAST-ITERATE CONVERGENCE FOR $\alpha_k = \frac{c}{k+1}$

**Recursion on $\mathbb{E}\big[\|e_{t,k}\|_2^2\big]$ :**    From Equation (38), we have that, whenever

$$k \ \geq \ k_{\mathrm{mix}} := \max\big(\tau_{\alpha_k},\, 2\tau_{\beta_k},\, \tau_\mu\big),$$

$$\mathbb{E}\big[\|e_{t,k+1}\|_2^2\big] \ \leq \ \Big(1 - \mu\alpha_k + C_e\,\alpha_k^2\Big)\,\mathbb{E}\big[\|e_{t,k}\|_2^2\big] \ + \ \alpha_k^2\,C_1 \ + \ \tfrac{4}{\mu}\,\alpha_k\,\big(\varepsilon_{\mathrm{approx}}^{\mathrm{dual}}\big)^2, \qquad \alpha_k = \tfrac{c}{k+1}. \tag{39}$$

Let

$$o_k \ := \ 1 - \mu\alpha_k + C_e\,\alpha_k^2, \qquad k_0 \ := \ \max\left\{\left\lceil\tfrac{2C_e c}{\mu}\right\rceil, \left\lceil\mu c\right\rceil\right\},$$

Then for all $j \geq k_0$, $0 < o_j \leq 1 - \frac{\mu c}{2}/(j+1) \leq 1$. When $k_{\mathrm{mix}} < k_0$, define the finite pre-burn product before the contraction kicks in as follows

$$H_{\mathrm{pre}} \ := \ \prod_{j=k_{\mathrm{mix}}}^{k_0-1} o_j, \qquad \text{and set } H_{\mathrm{pre}} = 1 \text{ if } k_{\mathrm{mix}} \geq k_0.$$

Assume the final iterate index $K > k_{\mathrm{mix}}$

**Tail-product bounds.**    For $m \leq u$, set $G_m^u := \prod_{j=m}^{u} o_j$. Then

$$G_{k_0}^{K-1} \ \leq \ \prod_{j=k_0}^{K-1}\Big(1 - \tfrac{\frac{\mu c}{2}}{j+1}\Big) \ \leq \ \left(\frac{k_0+1}{K}\right)^{\frac{\mu c}{2}}, \qquad \text{and if } k_{\mathrm{mix}} \geq k_0, \quad G_{k_{\mathrm{mix}}}^{K-1} \leq \left(\frac{k_{\mathrm{mix}}+1}{K}\right)^{\frac{\mu c}{2}}. \tag{40}$$

**Unrolling from $k_{\mathrm{mix}}$.**    Fix $K > k_{\mathrm{mix}}$. Unrolling equation 39 from $t = k_{\mathrm{mix}}$ to $K - 1$ yields

$$\mathbb{E}\big[\|e_{t,K}\|_2^2\big] \leq \underbrace{\mathbb{E}\big[\|e_{t,k_{\mathrm{mix}}}\|_2^2\big]\,G_{k_{\mathrm{mix}}}^{K-1}}_{\text{initial term}} + \sum_{t=k_{\mathrm{mix}}}^{K-1}\alpha_t^2 C_1\,G_{t+1}^{K-1} \ + \ \frac{4}{\mu}\big(\varepsilon_{\mathrm{approx}}^{\mathrm{dual}}\big)^2\sum_{t=k_{\mathrm{mix}}}^{K-1}\alpha_t\,G_{t+1}^{K-1}. \tag{41}$$

**Splitting at $k_0$ .**    Split each sum at $k_0$:

$$\sum_{t=k_{\mathrm{mix}}}^{K-1}(\cdot) \ = \ \underbrace{\sum_{t=k_{\mathrm{mix}}}^{k_0-1}(\cdot)}_{\text{finite ``pre-window''}} \ + \ \underbrace{\sum_{t=k_0}^{K-1}(\cdot)}_{\text{tail}}.$$

**Initial term.** If $k_{\mathrm{mix}} \geq k_0$, then by equation 40

$$\mathbb{E}\big[\|e_{t,k_{\mathrm{mix}}}\|_2^2\big]\,G_{k_{\mathrm{mix}}}^{K-1} \ \leq \ \mathbb{E}\big[\|e_{t,k_{\mathrm{mix}}}\|_2^2\big]\left(\frac{k_{\mathrm{mix}}+1}{K}\right)^{\frac{\mu c}{2}}.$$

If $k_{\mathrm{mix}} < k_0$, then

$$\mathbb{E}\big[\|e_{t,k_{\mathrm{mix}}}\|_2^2\big]\,G_{k_{\mathrm{mix}}}^{K-1} \ \leq \ \mathbb{E}\big[\|e_{t,k_{\mathrm{mix}}}\|_2^2\big]\,H_{\mathrm{pre}}\left(\frac{k_0+1}{K}\right)^{\frac{\mu c}{2}}.$$

**Variance sum.** Define the finite pre-window constant

$$U_{\mathrm{pre}}^{(\mathrm{mix})}(k_{\mathrm{mix}}, k_0) \ := \ \sum_{t=k_{\mathrm{mix}}}^{k_0-1}\frac{c^2 C_1}{(t+1)^2}\prod_{j=t+1}^{k_0-1} o_j \qquad (\text{define it as } 0 \text{ if } k_{\mathrm{mix}} \geq k_0).$$

Then

$$\sum_{t=k_{\text{mix}}}^{K-1} \frac{c^2 C_1}{(t+1)^2} G_{t+1}^{K-1} \leq U_{\text{pre}}^{(\text{mix})}(k_{\text{mix}}, k_0)\left(\frac{k_0+1}{K}\right)^{\frac{\mu c}{2}} + \sum_{t=k_0}^{K-1} \frac{c^2 C_1}{(t+1)^2}\left(\frac{t+2}{K}\right)^{\frac{\mu c}{2}}.$$

Using integral approximation, we can prove that the above term has the following upper bound:

$$\sum_{t=k_0}^{K-1} \frac{c^2 C_1}{(t+1)^2}\left(\frac{t+2}{K}\right)^{\frac{\mu c}{2}} \leq \begin{cases} \dfrac{c^2 C_1}{\frac{\mu c}{2}-1}\dfrac{1}{K} + \dfrac{c^2 C_1}{\frac{\mu c}{2}-1}\dfrac{(k_0+2)^{\frac{\mu c}{2}-1}}{K^{\frac{\mu c}{2}}}, & \frac{\mu c}{2} > 1, \\[2mm] \dfrac{c^2 C_1}{K}\left(1+\ln\dfrac{K}{k_0+2}\right), & \frac{\mu c}{2} = 1, \\[2mm] \dfrac{c^2 C_1}{1-\frac{\mu c}{2}}\dfrac{(k_0+2)^{\frac{\mu c}{2}-1}}{K^{\frac{\mu c}{2}}}, & 0 < \frac{\mu c}{2} < 1. \end{cases}$$

**Dual-approximation sum.** Define the finite pre-window constant

$$B_{\text{pre}}^{(\text{mix})}(k_{\text{mix}}, k_0) := \sum_{t=k_{\text{mix}}}^{k_0-1} \alpha_t \prod_{j=t+1}^{k_0-1} o_j \qquad (\text{again } 0 \text{ if } k_{\text{mix}} \geq k_0).$$

Then

$$\sum_{t=k_{\text{mix}}}^{K-1} \alpha_t\, G_{t+1}^{K-1} \leq B_{\text{pre}}^{(\text{mix})}(k_{\text{mix}}, k_0)\left(\frac{k_0+1}{K}\right)^{\frac{\mu c}{2}} + \sum_{t=k_0}^{K-1} \frac{c}{t+1}\left(\frac{t+2}{K}\right)^{\frac{\mu c}{2}}.$$

For the tail, an integral comparison yields

$$\sum_{t=k_0}^{K-1} \frac{c}{t+1}\left(\frac{t+2}{K}\right)^{\frac{\mu c}{2}} \leq \frac{c}{K^{\frac{\mu c}{2}}}\int_{k_0}^{K}(x+2)^{\frac{\mu c}{2}-1}dx = \frac{c}{\frac{\mu c}{2}K^{\mu c/2}}\left[(K+2)^{\mu c/2}-(k_0+2)^{\mu c/2}\right].$$

We also know that,

$$|\varepsilon_{\text{approx}}^{\text{dual}}| \leq \frac{1}{1-\gamma} + C_{F,M}.$$

We can also write,

$$\frac{c}{\frac{\mu c}{2}K^{\mu c/2}}\left[(K+2)^{\mu c/2}-(k_0+2)^{\mu c/2}\right]$$

$$\leq \frac{2}{\mu}\left(1+\frac{2}{K}\right)^{\frac{\mu c}{2}}$$

Therefore,

$$\frac{4}{\mu}\left(\varepsilon_{\text{approx}}^{\text{dual}}\right)^2 \sum_{t=k_{\text{mix}}}^{K-1} \alpha_t\, G_{t+1}^{K-1}$$

$$\leq \frac{4}{\mu}\left(\frac{1}{1-\gamma}+C_{F,M}\right)^2 B_{\text{pre}}^{(\text{mix})}(k_{\text{mix}}, k_0)\left(\frac{k_0+1}{K}\right)^{\frac{\mu c}{2}} + \frac{8}{\mu^2}\left(1+\frac{2}{K}\right)^{\frac{\mu c}{2}}(\varepsilon_{\text{approx}}^{\text{dual}})^2.$$

**Bounding the initial iterate at $k_{\text{mix}}$.** Recall that the slow update of Algorithm 1 satisfies

$$\theta_{t,k+1} = \theta_{t,k} + \alpha_k\, TD_{k+1}\,\phi(Z_{t,k}), \qquad \left\|TD_{k+1}\,\phi(Z_{t,k})\right\|_2 \leq 1 + C_{F,M} + \|\theta_{t,k}\|_2,$$

with $\theta_{t,0} = \mathbf{0}$. Define $u_k := \|\theta_{t,k}\|_2$. Then

$$u_{k+1} \leq u_k + \alpha_k\left(1 + C_{F,M} + u_k\right) = (1+\alpha_k)u_k + \alpha_k(1+C_{F,M}).$$

Introduce the shifted sequence $v_k := u_k + (1 + C_{F,M})$. We then have

$$v_{k+1} = u_{k+1} + (1 + C_{F,M}) \leq (1+\alpha_k)u_k + \alpha_k(1+C_{F,M}) + (1+C_{F,M})$$

$$= (1+\alpha_k)\left(u_k + 1 + C_{F,M}\right) = (1+\alpha_k)v_k.$$

Iterating from $k = 0$ to $k_{\mathrm{mix}} - 1$ and using $\theta_{t,0} = \mathbf{0}$ (so $u_0 = 0$ and $v_0 = 1 + C_{F,M}$) yields

$$v_{k_{\mathrm{mix}}} \;\leq\; (1 + C_{F,M}) \prod_{i=0}^{k_{\mathrm{mix}}-1} \left(1 + \alpha_i\right) = (1 + C_{F,M}) \prod_{i=0}^{k_{\mathrm{mix}}-1} \left(1 + \frac{c}{i+1}\right).$$

Using $1 + x \leq e^x$ and the harmonic-sum bound $\sum_{i=0}^{k_{\mathrm{mix}}-1} \frac{1}{i+1} \leq 1 + \ln k_{\mathrm{mix}}$, we obtain

$$\prod_{i=0}^{k_{\mathrm{mix}}-1} \left(1 + \frac{c}{i+1}\right) \;\leq\; \exp\!\left(\sum_{i=0}^{k_{\mathrm{mix}}-1} \frac{c}{i+1}\right) \;\leq\; \exp\!\left(c(1 + \ln k_{\mathrm{mix}})\right) = e^c\, k_{\mathrm{mix}}^c.$$

Therefore

$$u_{k_{\mathrm{mix}}} = \|\theta_{t,k_{\mathrm{mix}}}\|_2 \;\leq\; v_{k_{\mathrm{mix}}} \;\leq\; (1 + C_{F,M})\, e^c\, k_{\mathrm{mix}}^c.$$

Finally, recall that $e_{t,k} := \theta_{t,k} - \theta^{*,t}$, where $\theta^{*,t}$ is the (time-$t$) fixed point of the mean ODE. Using $\|e_{t,k_{\mathrm{mix}}}\|_2^2 \leq 2\|\theta_{t,k_{\mathrm{mix}}}\|_2^2 + 2\|\theta^{*,t}\|_2^2$ and the standard bound $\|\theta^{*,t}\|_2 \leq (1-\gamma)^{-1}/\sqrt{\mu}$, we obtain

$$\mathbb{E}\!\left[\|e_{t,k_{\mathrm{mix}}}\|_2^2\right] \leq 2(1 + C_{F,M})^2 e^{2c} k_{\mathrm{mix}}^{2c} \;+\; \frac{2}{\mu(1-\gamma)^2}$$

$$=: C_{\mathrm{init}}^{(\mathrm{mix},1)}.$$

This constant will be used as the initial-error term in the last-iterate bound for the $\alpha_k = c/(k+1)$ schedule.

**Final bound.** Combining the pieces, for any $K > k_{\mathrm{mix}}$,

$$\mathbb{E}\!\left[\|e_{t,K}\|_2^2\right]$$

$$\leq \begin{cases} C_{\mathrm{init}}^{(\mathrm{mix},1)} \left(\dfrac{k_{\mathrm{mix}}+1}{K}\right)^{\frac{\mu c}{2}}, & \text{if } k_{\mathrm{mix}} \geq k_0, \\[2ex] C_{\mathrm{init}}^{(\mathrm{mix},1)} H_{\mathrm{pre}} \left(\dfrac{k_0+1}{K}\right)^{\frac{\mu c}{2}}, & \text{if } k_{\mathrm{mix}} < k_0, \end{cases}$$

$$+ \; U_{\mathrm{pre}}^{(\mathrm{mix})}(k_{\mathrm{mix}}, k_0) \left(\tfrac{k_0+1}{K}\right)^{\frac{\mu c}{2}} \; + \; \begin{cases} \dfrac{c^2 C_1}{\frac{\mu c}{2}-1} \dfrac{1}{K} + \dfrac{c^2 C_1}{\frac{\mu c}{2}-1} \dfrac{(k_0+2)^{\frac{\mu c}{2}-1}}{K^{\frac{\mu c}{2}}}, & \frac{\mu c}{2} > 1, \\[2ex] \dfrac{c^2 C_1}{K} \left(1 + \ln \dfrac{K}{k_0+2}\right), & \frac{\mu c}{2} = 1, \\[2ex] \dfrac{c^2 C_1}{1 - \frac{\mu c}{2}} \dfrac{(k_0+2)^{\frac{\mu c}{2}-1}}{K^{\frac{\mu c}{2}}}, & 0 < \frac{\mu c}{2} < 1, \end{cases}$$

$$\frac{4}{\mu}\left(\frac{1}{1-\gamma} + C_{F,M}\right)^2 B_{\mathrm{pre}}^{(\mathrm{mix})}(k_{\mathrm{mix}}, k_0) \left(\frac{k_0+1}{K}\right)^{\frac{\mu c}{2}} \; + \; \frac{8}{\mu^2}\left(1 + \frac{2}{K}\right)^{\frac{\mu c}{2}} (\varepsilon_{\mathrm{approx}}^{\mathrm{dual}})^2.$$

## C.4 FORMAL VERSION OF THEOREM 1

**Theorem 2** (Main finite-time guarantee under function approximation). *Let $\hat{Q}_t := \Phi\hat{\theta}_t$ be the estimate of $Q^{\mathrm{rob},\pi}$ returned by Algorithm 1 at outer iteration $t$.*

$$k_{\mathrm{mix}}$$

$$:= \min\left\{ m \in \mathbb{N} : \forall j \geq m, \; j \;\geq\; \max\left(\tau_\mu, 2\left\lceil \frac{\log\!\left(\frac{C_{\mathrm{mix}}}{\beta_0}\sqrt{j+1}\right)}{\log(1/\rho)} \right\rceil, \left\lceil \frac{\log\!\left(C_{\mathrm{mix}}(j+1)/(c)\right)}{\log(1/\rho)} \right\rceil\right) \right\},$$

*where,*

$$\tau_\mu := \left\lceil \frac{\log\!\left(C_{\mathrm{mix}}\frac{1}{\mu}\right)}{\log(1/\rho)} \right\rceil$$

*Assume 1 holds, and we run $K \geq k_{\mathrm{mix}}$ inner iterations per outer iteration for either the TV or the Wasserstein-$\ell$ uncertainty sets.. Then, for any horizon $T \geq 1$,*

$$\mathbb{E}\big[\|\hat{Q}_T - Q^{\mathrm{rob},\pi}\|_\infty\big]$$

$$\leq \gamma^T \|\Phi\theta_0 - Q^{\mathrm{rob},\pi}\|_\infty \;+\; \frac{A_{\mathrm{sched}}(K)}{1-\gamma} \;+\; \frac{\epsilon_{\mathrm{approx}}}{1-\gamma} \;+\; \frac{2\sqrt{2}\left(1+\frac{2}{K}\right)^{\frac{\mu c}{4}}\epsilon_{\mathrm{approx}}^{\mathrm{dual}}}{\mu(1-\gamma)}. \qquad (42)$$

*Here $A_{\mathrm{sched}}(K) \geq 0$ is the schedule-dependent residual, which takes one of the following explicit forms depending on the range of c.*

*Recall the definition $k_0 = \max\left\{\left\lceil\frac{144c}{\mu}\right\rceil, \left\lceil\mu c\right\rceil\right\}$ and*

*Set*

$$o_j := \left(1 - \frac{c\mu}{j+1}\right) + \frac{72c^2}{(j+1)^2}, \quad H_{\mathrm{pre}} := \prod_{j=k_{\mathrm{mix}}}^{k_0-1} o_j,$$

$$U_{\mathrm{pre}}^{(\mathrm{mix})}(k_{\mathrm{mix}}, k_0) := \sum_{t=k_{\mathrm{mix}}}^{k_0-1} \frac{c^2 C_1}{(t+1)^2} \prod_{j=t+1}^{k_0-1} o_j, \quad B_{\mathrm{pre}}^{(\mathrm{mix})}(k_{\mathrm{mix}}, k_0) := \sum_{t=k_{\mathrm{mix}}}^{k_0-1} \alpha_t \prod_{j=t+1}^{k_0-1} o_j,$$

*with the convention that $H_{\mathrm{pre}} = 1$, $U_{\mathrm{pre}}^{(\mathrm{mix})} = 0$, $B_{\mathrm{pre}}^{(\mathrm{mix})} = 0$ when $k_{\mathrm{mix}} \geq k_0$. Also set*

$$C_{\mathrm{init}}^{(\mathrm{mix},1)} := 2(1 + C_{F,M})^2 e^{2c} k_{\mathrm{mix}}^{2c} \;+\; \frac{2}{\mu(1-\gamma)^2}.$$

*Then $A_{\mathrm{sched}}(K) = \sqrt{\Xi_1(K)}$ with*

$$\Xi_1(K) := \begin{cases} C_{\mathrm{init}}^{(\mathrm{mix},1)}\left(\dfrac{k_{\mathrm{mix}}+1}{K}\right)^{\frac{\mu c}{2}}, & k_{\mathrm{mix}} \geq k_0, \\[2ex] C_{\mathrm{init}}^{(\mathrm{mix},1)} H_{\mathrm{pre}}\left(\dfrac{k_0+1}{K}\right)^{\frac{\mu c}{2}}, & k_{\mathrm{mix}} < k_0, \end{cases}$$

$$+ \; U_{\mathrm{pre}}^{(\mathrm{mix})}(k_{\mathrm{mix}}, k_0)\left(\frac{k_0+1}{K}\right)^{\frac{\mu c}{2}}$$

$$+ \begin{cases} \dfrac{c^2}{\frac{\mu c}{2}-1}\dfrac{1}{K}C_1 \;+\; \dfrac{c^2}{\frac{\mu c}{2}-1}\dfrac{(k_0+2)^{\frac{\mu c}{2}-1}}{K^{\frac{\mu c}{2}}}C_1, & \frac{\mu c}{2} > 1, \\[2ex] \dfrac{c^2}{K}\left(1 + \ln\dfrac{K}{k_0+2}\right)C_1, & \frac{\mu c}{2} = 1, \\[2ex] \dfrac{c^2}{1-\frac{\mu c}{2}}\dfrac{(k_0+2)^{\frac{\mu c}{2}-1}}{K^{\frac{\mu c}{2}}}C_1, & 0 < \frac{\mu c}{2} < 1, \end{cases}$$

$$+ \; \frac{4}{\mu}\left(\frac{1}{1-\gamma} + C_{F,M}\right)^2 B_{\mathrm{pre}}^{(\mathrm{mix})}(k_{\mathrm{mix}}, k_0)\left(\frac{k_0+1}{K}\right)^{\frac{\mu c}{2}}.$$

**Restating the Constants**

$$C_0 = 32\gamma^2 C_{F,M}^2 + 32\left(1 + \gamma C_{F,M} + \frac{1}{(1-\gamma)\sqrt{\mu}}\right)^2,$$

$$C_{\mathrm{cross}} = 8\left(1 + \gamma C_{F,M} + \frac{1}{(1-\gamma)\sqrt{\mu}}\right)^2,$$

$$C_{\mathrm{fast}} = \frac{4B_\nu^2 + \beta_0^2(8B_\nu C_G + 25C_G^2)\ln(2e)}{\beta_0},$$

$$C_{\mathrm{bias}} = C_{\mathrm{fast}}^2 \;+\; 2C_{F,M}^2 \;+\; 64\,C_{F,M}^2\,\frac{C_{\mathrm{mix}}\rho}{1-\rho},$$

$$C_1 = \left(\left(\tfrac{8}{c\mu}+4\right)C_{bias} + C_{cross} + C_0 + 4\left(\frac{1}{1-\gamma}+C_{F,M}\right)^2\right).$$

## D  REMAINING PROOFS

### D.1  BOUND ON $\theta_t^*$

We drop the superscript $t$ from $\theta_t^*$ as $t$ is fixed throught the discussion of this subsection. Recall,

$$\theta_t^* = (\Phi^\top D^\pi \Phi)^{-1} \Phi^\top D^\pi [r + \gamma \bar{f}_t(\lambda^*)]$$

**Lemma 7** (Bound on the optimal weight vector). *Let*

$$\theta^* = \left(\Phi^\top D^\pi \Phi\right)^{-1} \Phi^\top D^\pi \left[r + \gamma \bar{f}_t(\lambda^*)\right],$$

*where*

- $\Phi \in \mathbb{R}^{|\mathcal{S}||\mathcal{A}| \times d}$ *has full column rank and row vectors* $\phi(s, a)$ *satisfying* $\|\phi(s, a)\|_2 \leq 1$;

- $\Phi^\top D^\pi \Phi$ *is positive definite with min-eigenvalue* $\nu$. *The diagonal entries of the matrix* $D^\pi$ *satisfy* $d_i \geq d_{\min} > 0$ *and* $\sum_i d_i = 1$;

- *each entry of* $r$ *obeys* $|r_i| \leq 1$;

- *each entry of* $\bar{f}_t(\lambda^*)$ *obeys* $\left|\bar{f}_t(\lambda^*)_i\right| \leq 1/(1-\gamma)$.

*Then*

$$\|\theta^*\|_2 \leq \frac{1}{1-\gamma} \frac{1}{\sqrt{\mu}}.$$

*Proof.* Set

$$C := \Phi^\top D^\pi \Phi, \qquad v := r + \gamma \bar{f}_t(\lambda^*).$$

By definition of $\theta^*$,

$$C\theta^* = \Phi^\top D^\pi v.$$

Multiply by $\theta^{*\top}$ on the left:

$$\theta^{*\top} C\theta^* = \theta^{*\top} \Phi^\top D^\pi v = (\Phi\theta^*)^\top D^\pi v.$$

Let $y := \Phi\theta^*$. Then

$$\theta^{*\top} C\theta^* = y^\top D^\pi v.$$

On the other hand,

$$\theta^{*\top} C\theta^* = \theta^{*\top} \Phi^\top D^\pi \Phi\theta^* = (\Phi\theta^*)^\top D^\pi (\Phi\theta^*) = y^\top D^\pi y.$$

Thus

$$y^\top D^\pi y = y^\top D^\pi v.$$

Apply Cauchy–Schwarz in the $D^\pi$–weighted inner product:

$$y^\top D^\pi v = (D^{\pi 1/2} y)^\top (D^{\pi 1/2} v) \leq \|D^{\pi 1/2} y\|_2 \|D^{\pi 1/2} v\|_2 = (y^\top D^\pi y)^{1/2} (v^\top D^\pi v)^{1/2}.$$

If $y^\top D^\pi y = 0$, then $\theta^* = 0$ and the desired bound is trivial, so assume $y^\top D^\pi y > 0$ and divide both sides by $(y^\top D^\pi y)^{1/2}$:

$$(y^\top D^\pi y)^{1/2} \leq (v^\top D^\pi v)^{1/2} \implies y^\top D^\pi y \leq v^\top D^\pi v.$$

Recalling $y^\top D^\pi y = \theta^{*\top} C\theta^*$, we obtain

$$\theta^{*\top} C\theta^* \leq v^\top D^\pi v.$$

**Upper bound on** $v^\top D^\pi v$**.** For each component,

$$|v_i| \leq |r_i| + \gamma|\bar{f}_t(\lambda^*)_i| \leq 1 + \frac{\gamma}{1-\gamma} = \frac{1}{1-\gamma}.$$

Hence

$$v^\top D^\pi v = \sum_i d_i v_i^2 \leq \sum_i d_i \left(\frac{1}{1-\gamma}\right)^2 = \left(\frac{1}{1-\gamma}\right)^2.$$

**Lower bound via the minimum eigenvalue.** Since $C = \Phi^\top D^\pi \Phi \succeq \mu I$,

$$\theta^{*\top} C \theta^* \ \geq \ \mu \|\theta^*\|_2^2.$$

Combining the upper and lower bounds,

$$\mu \|\theta^*\|_2^2 \ \leq \ \theta^{*\top} C \theta^* \ \leq \ \Big(\frac{1}{1-\gamma}\Big)^2,$$

so

$$\|\theta^*\|_2 \ \leq \ \frac{1}{1-\gamma}\frac{1}{\sqrt{\mu}}.$$

$\square$

## E  ROBUST Q-LEARNING

In this section, we discuss a robust Q-learning algorithm with function approximation that finds the optimal policy for the worst-case transition kernel in the uncertainty set considered in this paper. We first define the optimal state-action value function $Q^{\mathrm{rob},*}$ as the state-action value function of the best admissible policy to maximize $Q^{\mathrm{rob},\pi}$ for each $(s,a)$-pair.

$$Q^{\mathrm{rob},*}(s,a) = \max_\pi Q^{\mathrm{rob},\pi}(s,a), \forall (s,a) \in \mathcal{S} \times \mathcal{A}.$$

It is shown in prior literature (Iyengar, 2005) that $Q^{\mathrm{rob},*}$ satisfies the following equation, which is called the robust Bellman optimality equation

$$Q^{\mathrm{rob},*}(s,a) \ = \ R(s,a) \ + \ \gamma \min_{q \in \mathcal{P}_s^a} \sum_{s'} q(s') \underbrace{\max_{a'} Q^{\mathrm{rob},*}(s',a')}_{=: \ V^{\mathrm{rob},*}(s')}. \tag{43}$$

Equivalently, define the robust Bellman optimality operator $(\mathcal{T}^{\mathrm{rob},*} Q)(s,a) \ := \ R(s,a) \ + \ \gamma \, \sigma_{\mathcal{P}_s^a}(V^{Q,*})$ with

$$V^{Q,*}(s') := \max_{a'} Q(s',a'), \tag{44}$$

and $\sigma_{\mathcal{P}_s^a}(V)$ is given in Equation (4). Iyengar (2005) proved that the robust Bellman optimality operator is $\gamma$-contraction in $\ell_\infty$ norm.

Now, we discuss how the TD learning algorithm presented in Algorithm 1 in the main body of the paper can be extended to estimate $Q^{\mathrm{rob},*}$ in a relatively straightforward manner.Similar to the TD learning setup, assume that we can sample data corresponding to a behavioral policy $\pi_b$ from the nominal model $P_0$. Also, assume that the policy $\pi_b$ satisfies Assumption 1.

The goal here is to approximate $Q^{\mathrm{rob},*}$ by $\Phi \theta^*$ for an appropriately chosen $\theta^*$. Our Q-learning algorithm is presented in Algorithm 2. The algorithm computes an estimate $\hat{\theta}_t$ of this parameter at each iteration $t$ of the outer loop. The quantity $V_{\hat{\theta}_t}^{\mathrm{rob},*}$ in the description of the algorithm is given by

$$V_{\hat{\theta}_t}^{\mathrm{rob},*}(s) = \max_a \mathrm{Clip}\left(\phi(s,a)^\top \hat{\theta}_t\right), \forall s \in \mathcal{S}. \tag{45}$$

**Difference between Algorithm 2 and Algorithm 1:** The only difference between the robust Q-learning algorithm in Algorithm 2 and the robust TD learning algorithm in Algorithm 1 is that, we use $V_{\hat{\theta}_t}^{\mathrm{rob},*}$ instead of $V_{\hat{\theta}_t}^{\mathrm{rob}}$ in the calculation of the dual super-gradient in line 6 and the calculation of the dual objective in line 10 in Algorithm 2.

**Finite-Time Performance Bound for the Robust Q-Learning (Algorithm 2):** Recall that we established a finite-time performance bound for the robust TD learning in Theorem 1. By following the steps of the proof of that theorem, it is easy to see that an analogous guarantee holds for the estimate of $Q^{\mathrm{rob},*}$ produced by Algorithm 2. The reason that the proof is identical is that the robust Bellman optimality operator is a $\gamma$-contraction in the $\ell_\infty$ norm as was the robust Bellman operator

for a fixed policy. The only difference is that the function approximation error for approximating the Q-function should now be defined as the error in approximating $Q^{\mathrm{rob},*}$ by the class of functions $\{\Phi\theta : \theta \in \mathbb{R}^{n_\theta}\}$ :

$$\epsilon^*_{\mathrm{approx}} := \sup_{Q=\mathrm{Clip}(\Phi\theta),\theta\in\mathbb{R}^{n_\theta}} \left\| \mathrm{Clip}\left(\Pi\mathcal{T}^{\mathrm{rob},*}(Q)\right) - \mathcal{T}^{\mathrm{rob},*}(Q) \right\|_\infty. \tag{46}$$

The above definition is completely analogous to the TD learning setting in the main body of the paper, but with $\mathcal{T}^{\mathrm{rob},*}$ instead of $\mathcal{T}^{\mathrm{rob},\pi}$ for a policy $\pi$.

Thus, the sample complexity of robust Q-learning is of the same order as that of robust TD-learning up to a function approximation error.

---

**Algorithm 2** Robust Q-learning with Function Approximation

---

1: **Input:** Integers $T, K$. Initial $\nu_0 \in \mathbb{R}^{n_\lambda}$, $\theta_0 :=$ zero vector, fast time-scale step-sizes $\beta_k = \frac{\beta_0}{\sqrt{k+1}}$, slow time-scale step-sizes $\alpha_k = \frac{c}{(k+1)}$ for some $c : 0 < c < \infty$; $\hat{\theta}_0 = \theta_0, \theta_{0,0} = \theta_0$, behavioral policy $\pi_b$, Reward function $R : (\mathcal{S} \times \mathcal{A}) \mapsto [-1, 1]$, initial state $S_{0,0}$.
2: **for** $t = 0, 1, \cdots, T - 1$ **do**
3:     **for** $k = 0, 1, \ldots, K - 1$ **do**
4:         Take action $A_{t,k}$ according to the behavioral policy $\pi_b$ and sample $S_{t,k+1}$ ($S_{t,k+1} \sim P_0(\cdot|S_{t,k}, A_{t,k})$)
5:     **fast time-scale** $(\beta_k)$
6:         Compute $\hat{G}(\psi(S_{t,k}, A_{t,k})^\top \nu_{t,k}; V^{\mathrm{rob},*}_{\hat{\theta}_t}, S_{t,k+1})$ from Equation (17) for TV uncertainty set and Equation (20) for Wasserstein-$\ell$ uncertainty set
7:         $\nu_{t,k+1} = \mathrm{Proj}_{\mathcal{M}_\nu}(\nu_{t,k} + \beta_k[\hat{G}(\psi(S_{t,k}, A_{t,k})^\top \nu_{t,k}; V^{\mathrm{rob},*}_{\hat{\theta}_t}, S_{t,k+1})\psi(S_{t,k}, A_{t,k})])$
8:     **Slow time-scale** $(\alpha_k)$
9:         Compute $\bar{\nu}_{t,k}$ from Equation (7)
10:        Compute $\hat{F}(\psi(S_{t,k}, A_{t,k})^\top \bar{\nu}_{t,k}; V^{\mathrm{rob},*}_{\hat{\theta}_t}, S_{t,k+1})$ from Equation (18) for TV uncertainty set and Equation (21) for Wasserstein-$\ell$ uncertainty set
11:        $TD_{t,k+1} = R(S_{t,k}, A_{t,k}) + \gamma\hat{F}(\psi(S_{t,k}, A_{t,k})^\top \bar{\nu}_{t,k}; V^{\mathrm{rob},*}_{\hat{\theta}_t}, S_{t,k+1}) - \phi(S_{t,k}, A_{t,k})^\top \theta_{t,k}$
12:        $\theta_{t,k+1} = \theta_{t,k} + \alpha_k TD_{t,k+1}\phi(S_{t,k}, A_{t,k})$
13:     **end for**
14:     $\hat{\theta}_{t+1} = \theta_{t,K}, S_{t+1,0} = S_{t,K}, \theta_{t+1,0} = \theta_{t,K}, \nu_{t+1,0} = \nu_{t,K}$.
15: **end for**
16: **Output:** $\hat{\theta}_T$

---

## USE OF LARGE LANGUAGE MODEL

The authors used large language models (e.g., ChatGPT) to polish the language in certain parts of the paper. All technical content, proofs, and conclusions are the sole work of the authors.

