# OpenReview forum: "Finite‑Time Bounds for Distributionally Robust TD Learning with Linear Function Approximation"
_ICLR.cc/2026/Conference — Submitted to ICLR 2026_

### Official Review · Reviewer_Qgkf · 2025-10-25

**Soundness:** 3
**Presentation:** 2
**Contribution:** 3
**Rating:** 6
**Confidence:** 4

**Summary:**

This work provides a non-generative robust TD algorithm and establishes its finite-time convergence bounds. It analyzes the required sample size under both total variation and Wasserstein distances. Notably, the convergence of the proposed robust TD algorithm does not rely on strict contraction assumptions. Overall, this work introduces a novel algorithm and offers a fundamental understanding of distributionally robust reinforcement learning (DRRL).

**Strengths:**

1.  This work provides the robust TD algorithm, and firstly does not rely on extra assumptions within the TV and Wasserstein uncertainty set.

2. This work provides solid theoretical result.

**Weaknesses:**

1. The work approximates the Lagrange multipliers using linear function approximation. This raises a concern regarding whether the assumption is reasonable and whether the resulting approximation error can be sufficiently small, given that the Lagrange multipliers induce a nonlinear correction within the Wasserstein uncertainty set.

2. Is the parameter $\mu$ sufficiently small? The final result contains the term $\epsilon_{\text{approx}} / \mu$, which suggests potential instability or sensitivity to the choice of $\mu$.

3. There appears to be a typo in the definition of $G$ — the last line should use $G$ instead of $g$.

4. The definitions and assumptions in equations (19), (20), (22), and (23) in the appendix seem unreasonable or inconsistent with the main formulation. Further clarification or justification would be helpful.

**Questions:**

Please check the Weaknesses part.

**Details Of Ethics Concerns:**

NaN

---

> ### Author Response · Authors · 2025-11-23
> **Response**
>
> **Comment:**
> The work approximates the Lagrange multipliers using linear function approximation. This raises a concern regarding whether the assumption is reasonable and whether the resulting approximation error can be sufficiently small, given that the Lagrange multipliers induce a nonlinear correction within the Wasserstein uncertainty set.
>
> **Response:**
> Regarding the dual function approximation error, this is exactly the same modeling choice used throughout in the RL theory literature, where linear function approximation is routinely used for complex objects such as value functions, and the residual approximation error is explicitly captured in the final guarantees. Our paper provides a theoretical convergence guarantee for DRRL under linear function approximation, while nonlinear function approximation is already used empirically in similar algorithms (e.g., [1]); from a theory perspective, this is essentially as far as the state of the art with function approximation goes, even in the non-robust setting. Moreover, wide neural networks are known to admit linearized descriptions (e.g., via the neural tangent kernel), which further supports the relevance of studying linear function approximation for the dual variables.
>
> **Comment:**
> Is the parameter $\mu$ sufficiently small? The final result contains the term $\frac{\epsilon_{approx}}{\mu}$ , which suggests potential instability or sensitivity to the choice of $\mu$.
>
> **Response:**
> Here $\mu$ is the minimum eigenvalue of $\Phi^\top D \Phi$, assumed positive—a standard assumption that captures how well-conditioned the feature representation is. The term $\epsilon^{dual }_{approx} / \mu$ reflects that the convergence error worsens when the features are nearly linearly dependent (i.e., when the problem is ill-conditioned), which is already known to be unavoidable in non-robust RL.
>
> In practice, we typically do not hand-design feature vectors; instead, neural networks are used to also find feature vectors. Empirically, \cite{liang2023single} shows that neural networks can approximate both the dual variables and the value functions quite effectively in similar DRRL implementations.
>
>
> **Comment:**
> There appears to be a typo in the definition of  $G$— the last line should use $G$ instead of $g$.
>
>
> **Response:**
> Thanks for pointing it out; it is fixed in the uploaded revised version.
>
> **Comment:** The definitions and assumptions in equations (19), (20), (22), and (23) in the appendix seem unreasonable or inconsistent with the main formulation. Further clarification or justification would be helpful.
>
> **Response:**
>
> Thanks for pointing this out. The inconsistencies in notations and definitions have been removed in the uploaded revised version.
>
> [1] Zhipeng Liang, Xiaoteng Ma, Jose Blanchet, Jiheng Zhang, and Zhengyuan Zhou. Single-trajectory
> distributionally robust reinforcement learning. arXiv preprint arXiv:2301.11721, 2023.

---

### Official Review · Reviewer_FRHC · 2025-10-31

**Soundness:** 3
**Presentation:** 2
**Contribution:** 3
**Rating:** 4
**Confidence:** 4

**Summary:**

This paper derives a finite-time bound for distributionally robust TD-learning with linear function approximation. It primarily focus on the robustness with respect to the transition probability; that is, it pre-assumes the transition probability falls into a given set, named uncertainty set, and it considers the worst-case value function among the given uncertainty set. The linear feature is adopted to approximate the Q-function. The goal is to learn the robust Q-function under this setting. A general assumption is given to characterize the interested uncertainty set; two well-known uncertainty sets, wasserstein and total varioation uncertainty sets satisfy the given condition. Then the convergence guarantee is given based on the scheduler.

**Strengths:**

This paper is addressing an important problem in robust RL; that is, how to accurately and efficiently solve the robust value function. The presented results indicate that under the linear approximation (with some traditional requirements), the robust TD-learning will finally learn the robust value function in the desired complexity.

 The theoretical analysis sounds to me as it matches the best-known complexity for the non-robust RL literature under the approperiate parameter setting. All related assumptions to guarantee the convergence are also used in non-robust RL literature. Taking the target network  also introduces a new technique in robust RL.

**Weaknesses:**

1. Turnning the optimization problem into a distributionally robust optimization problem and applies the dual form is not new. Assumption 1 simply says that this optimization problem can be solvable in the desired complexity. It typically cannot be considered as a novel contribution.

2. The presentation is not very clear:
    1. Many notations are used without defining them or being defined somewhere hard to find. For example, $\mu$ and $C_e$ in Line 352.
    2. And the complete form of $C^\star$, $C_1$, and $C_2$ are not very helpful here. And the author doesn't elaborate anything on them.
    3. In Line 390, "it requires c to be chosen sufficiently large". But it is unclear why it is an issue; is there any problem for using  $1/\mu$?
    4. Some statements do not have explainations; see my questions below.

3. No experiments validating the necessasity of the two-time scale.

4. Writing issues:  The author commonly omitts punctuation marks in formulas.  Many LaTeX issues.

**Questions:**

1. Can the proposed Assumption 1 motivate other uncertainty sets beyond the total variation and Wasserstein? As those distances have been well-studied (e.g. the total variation has been covered by the IPM from *Natural Actor-Critic for Robust Reinforcement Learning with Function Approximation* under linear function approximation), it is hard to find how useful the proposed concept is; it just sounds like for convenience of analysis.

2. The author commented that "Prior work Zhou etal.(2023) circumvents this by imposing restrictive assumptions on $\gamma$". I didn't see their restrictive assumptions. Can the author further clarify it?

3. How should I understand the complexity calculated in Corollary 1? Has the author included the proof somewhere?

4. Can author include some experiments? It is hard to tell if the two-time scale is necessary or more efficient. It would be better to
    1. Compare it with the single-time scale baseines.
    2. Compare with robust TD-learning in IPM uncertainty sets used in  Zhou et al.(2023)  *Natural Actor-Critic for Robust Reinforcement Learning with Function Approximation*.
    3. Validate the compatibility of this approach with policy gradient or other policy gradient based algorthms.

---

> ### Author Response · Authors · 2025-11-23
> **Response, Part 1**
>
> **Comment:** Turning the optimization problem into a distributionally robust optimization problem and applies the dual form is not new. Assumption 1 simply says that this optimization problem can be solvable in the desired complexity. It typically cannot be considered as a novel contribution.
>
> **Response**
> We agree that the dual form idea for distributionally robust problems is not new and we have acknowledge this fact in the paper. As stated in the paper, these ideas go back to [1]. Moreover, in DRRL, the idea is used in the theory for the tabular setting and in neural network-based function approximation simulations in [2]. Our key contribution is to combine the analytical approaches from different domains in a non-trivial manner: target network approach used in non-robust Q-learning to deal with contraction issue, distributionally robust optimization to work in the dual form of the robust RL problem and multi-time scale stochastic approximation to tackle the problem of biased single-sample estimator of the inner optimization problem in DRRL. Each of these analytical techniques and the corresponding convergence have been previously studied in isolation; making these tools work together in the DRRL setting with function approximation is non-trivial. In particular, we need to carefully track their interactions so that the resulting convergence rate yields a sample complexity that is order-wise the same as in the non-robust case, with the same dependencies on $\gamma$.
>
> Here is an example of the non-trivial work that was needed to get the rate of convergence that we were able to obtain. Function approximation in the dual space and the estimate of the dual variables introduce both a bias and a variance in the estimate of the value function. So both have to be controlled appropriately. To study the bias, we use the average of the suboptimality gap in the dual objective at each (state, action) pair weighted by the stationary probability of visiting the (state, action) pair. One could potentially use other metrics to measure the accuracy of the dual variable estimation, but this particular metric is crucial for the proofs to work out. To control the variance, we use averaging of the dual parameter vector (line 9 in the algorithms) in the algorithms, which to the best of our knowledge, had not been used previously in this context. Without averaging, the rate of convergence does not match the non-robust case.
>
> **Comment:** The presentation is not very clear: ...
>
>
> **Response:**
> We thank the reviewer for pointing out the typos and inconsistencies in the submitted version of the paper. We have uploaded a revised version after carefully fixing all inconsistency issues in the draft.
>
> In the revised version, the complete form of the constants is deferred to the appendix and the dependency of the sample complexity on the problem parameter $\gamma$ is explicitly stated in the main body to improve readability. We note that the $\gamma$ dependency in our result is in line with the results in non-robust Q-learning in prior literature (\cite{chen2023target}).
>
> We do not know the nominal model we only have access to Markovian data from that model. Hence, we do not know $\mu$. Nevertheless, our theorem establishes finite-time convergence guarantee even when $c$ is smaller than $\frac{1}{\mu}$. We argue in Line 390 that, the best sample complexity is achieved if we have sufficiently large $c$. {This observation is in line with prior works in non-robust RL such as [3].
>
>
>
>
>
>
> [1] Garud N Iyengar. Robust dynamic programming. Mathematics of Operations Research, 30(2):
> 257–280, 2005.
>
> [2] Zhipeng Liang, Xiaoteng Ma, Jose Blanchet, Jiheng Zhang, and Zhengyuan Zhou. Single-trajectory
> distributionally robust reinforcement learning. arXiv preprint arXiv:2301.11721, 2023.
>
> [3] Zaiwei Chen, John-Paul Clarke, and Siva Theja Maguluri. Target network and truncation overcome
> the deadly triad in-learning. SIAM Journal on Mathematics of Data Science, 5(4):1078–1101,
> 2023.
>
> [4] Ruida Zhou, Tao Liu, Min Cheng, Dileep Kalathil, PR Kumar, and Chao Tian. Natural actor-critic
> for robust reinforcement learning with function approximation. Advances in neural information
> processing systems, 36:97–133, 2023.

---

> ### Author Response · Authors · 2025-11-23
> **Response, Part 2**
>
> **Comment:** Can the proposed Assumption 1 motivate other uncertainty sets beyond the total variation and Wasserstein? As those distances have been well-studied (e.g. the total variation has been covered by the IPM from Natural Actor-Critic for Robust Reinforcement Learning with Function Approximation under linear function approximation), it is hard to find how useful the proposed concept is; it just sounds like for convenience of analysis.
>
> **Response**
>
> In the revised version of the paper, we have moved Assumption 1 to the appendix and stated the theorem for TV and Wasserstein distances. In the new version, Assumption 1 is simply a device to unify the proofs for these distances. One can extend Assumption 1 in a manner such that it includes the Cressie-Read family, but since we have not done that yet, we have also reworded our contributions.
>
> The work in [4] designs a specific form of IPM distance metric specifically tailored to their function-approximation class and does not include TV distance. Moreover, they rely on a very restrictive condition on the discount factor (discussed in detail in our response to the next comment).
>
> **Comment:** The author commented that "Prior work Zhou etal.(2023) circumvents this by imposing restrictive assumptions on ". I didn't see their restrictive assumptions. Can the author further clarify it?
>
> **Response:**
> The restrictive assumption is Assumption 2 in the paper [4]. The assumption says that there exists a $\beta \in (0,1)$ such that $\gamma \leq \frac{\beta P_0(s'|s,a)}{P(s'|s,a)}$ for all $s,s' \in \mathcal{S}$, $a \in \mathcal{A}$ and for all $P$ from the uncertainty set. Let us take an example to parse the assumption. Let $P_0(\cdot|s,a)$ for some $(s,a)$ be $(0.1,0.2,0.3,0.4)$ and let us consider TV distance with radius $0.5$. Then, $P(\cdot|s,a) \coloneqq (0.6,0.2,0.1,0.1)$ lies in the uncertainty set. This example requires $\gamma \leq \frac{1}{6}$. Hence, this condition forces $\gamma$ to be very small.
>
> **Comment:** How should I understand the complexity calculated in Corollary 1? Has the author included the proof somewhere?
>
> **Response:** We thank the reviewer for pointing this out. In the revised draft, we added the derivation of the sample complexity before stating the corollary. The Corollary follows immediately from the statements of the Theorem in the main body of
> the paper.
>
> **Comment:** Can author include some experiments? ...
>
> **Response:** The contribution of this paper is theoretical. As mentioned in the paper, there have been prior works with empirical validation of robust RL but the theory in those papers are very different from the practical implementations. Our work is the first to address function approximation (without extremely restrictive assumptions) in DRRL.
>
> **Comment** Writing issues..
>
> **Response** We thank the reviewer for pointing out writing issues like punctuation or latex issues. We uploaded a revised version where we fixed these issues to the best of our knowledge.
>
>
> [4] Ruida Zhou, Tao Liu, Min Cheng, Dileep Kalathil, PR Kumar, and Chao Tian. Natural actor-critic for robust reinforcement learning with function approximation. Advances in neural information processing systems, 36:97–133, 2023.

---

> > ### Comment · Reviewer_FRHC · 2025-11-23
> >
> > > Comment: The author commented that "Prior work Zhou etal.(2023) circumvents this by imposing restrictive assumptions on ". I didn't see their restrictive assumptions. Can the author further clarify it?
> > >
> > > Response: The restrictive assumption is Assumption 2 in the paper [4].
> >
> > A quick comment: This statement is not exactly true. [4] exactly states that this assumption is not necessary. In [4], they are using the  IPM uncertainty set, and it doesn't rely on this assumption in their theoretical result.
> >
> > And they have already addressed the linear approximation setting. The TV distance is a special case of the IPM. From this point, I may not agree that the current submission is "the first to address function approximation (without extremely restrictive assumptions) in DRRL."

---

> ### Author Response · Authors · 2025-11-23
> **Response to the Comment on Limitations of Prior Work Zhou et al (2023)**
>
> We respectfully maintain that our characterization of the prior work [4] (https://arxiv.org/pdf/2307.08875) as relying on restrictive conditions and not including the TV distance uncertainty set is accurate.
>
> [4 ]notes on page 6 in their paper that Assumption 2 may not be necessary for their IPM-based uncertainty set, and that contraction of the projected robust Bellman operator can instead be ensured under the condition in their Lemma 1. In the proof of Theorem 5 (page 23), they indeed invoke Lemma 1 to guarantee contraction in the robust policy-evaluation step. **Our point is that the alternative condition in Lemma 1 is itself restrictive**. Specifically, Lemma~1 requires
>
> $$
> \delta \leq   \lambda_{\min}(\Psi^\top D^\pi \Psi) \frac{1-\gamma}{\gamma},
> $$
>
> where $\delta$ is the uncertainty radius and $\lambda_{\min}(\Psi^\top D^\pi \Psi)$ is the minimum eigenvalue of the matrix $\Psi^\top D^\pi \Psi$. Rearranging gives
> $$
> \gamma \leq  \frac{\lambda_{\min}(\Psi^\top D^\pi \Psi)}{\lambda_{\min}(\Psi^\top D^\pi \Psi)+\delta}.
> $$
> For normalized features, $\lambda_{\min}(\Psi^\top D^\pi \Psi)$ is at most $1$, and in practice can be much smaller when the features are poorly conditioned. For a nontrivial robustness radius $\delta$, the ratio above can then be significantly small. Moreover, scaling the features to increase $\lambda_{\min}$ also scales the radius $\delta$ through the function class defining the IPM (ref. their Equation (5)), so this issue cannot be removed by simple rescaling.
>
> Regarding TV distance: while TV can be written as a special case of an IPM with a particular function class, **the specific IPM considered in [4] (Equation (5)) does not correspond to TV**. Their proof of Lemma 1 (page 22) crucially uses the property that, under their chosen IPM, one can construct an unbiased single-sample estimator of the robust Bellman operator based on the next transition. This property relies on the structure of their function class and does not hold for TV balls, where, to the best of our knowledge (and as discussed in Section~2.1 of our paper), no such unbiased single-sample estimator is available. Thus, their analysis does not cover TV (or Wasserstein) ambiguity sets under linear function approximation.
>
> [4] Ruida Zhou, Tao Liu, Min Cheng, Dileep Kalathil, PR Kumar, and Chao Tian. Natural actor-critic for robust reinforcement learning with function approximation. Advances in neural information processing systems, 36:97–133, 2023;
> Link to the paper: https://arxiv.org/pdf/2307.08875

---

> > ### Comment · Reviewer_FRHC · 2025-11-26
> >
> > Thanks for the response.
> >
> > In general, I am satisfied with the clarification on the novelty and the improved presentation. I also agree that [4] requires a specific function class and the IPM cannot include the Wasserstein case.
> >
> > Concerns not addressed:
> >
> > (a) I disagree with the argument that the alternative assumption used in the robust natural actor-critic paper is too strong. It simply requires to let the radius to be sufficiently small. This parameter can be determined at the beginning of training. And this work didn't completely remove this assumption; it introduces another assumption "the weighted feature covariance is well-conditioned" to make the robust Bellman operator a contraction.
> >
> > (b) I also disagree that the IPM cannot include the total variation. The author claims in Sec.2.1 that "Except for special uncertainty sets (e.g., R-contamination), there is no direct plug-in unbiased single-sample estimator of this inner minimum, which creates a bias in standard TD updates". But it is not a proof; it is just an intuitive explaination.
> >
> > (c) This work proposes a new algorithm: use the two-time scale stochastic approximation to address the challenge "no direct plug-in unbiased single-sample estimator". Without any empirical evidence, it is hard to know if this method really addresses this challenge as claimed in the paper (on page 4, Line 191-Line 199).
> >
> > ===
> >
> > For these reasons, I cannot support for accepting this paper. But I would be fine if the AC rules over this decision.

---

> ### Author Response · Authors · 2025-11-26
> **Clarification on Prior Assumptions, TV/IPM Coverage, and Our Contribution**
>
> We respectfully disagree and believe the reviewer has incorrectly understood the results in prior work and the assumption in our work. We have provided our response below and would be happy to engage in further discussions.
>
> **(a)** There appears to be confusion between two different issues. One is that the paper by [4] uses a restrictive condition on $\gamma$ depending on the minimum eigenvalue (which we denote by $\mu$ in our paper). Our paper removes such an assumption on $\gamma$. The second is the dependence of the convergence of RL algorithms on $\mu$, which is well known even in the non-robust RL literature ([1],[2],[3]). We do not impose any well-conditionedness assumption: others and we point out that the rate of convergence of TD learning and Q-learning depend on how well conditioned the feature vectors are. This has nothing to do with $\gamma$ and is not an assumption.
>
> **(b)** We are not saying that IPMs cannot include TV distance. In fact, the TV distance is an IPM. Our point is that the paper [4], while starting from a general IPM framework, uses a specific IPM which does not include the TV uncertainty set. The specific IPM they use is defined via a function class that does not correspond to TV distance. The primal robust optimization problem is
>
> $$
> \sigma_{\mathcal{P}_s^a}(V) = min _{ q \in \mathcal{P}_s^a} \sum _{s'} q(s') V(s').
> $$
>
> When using TV or Wasserstein-$\ell$ uncertainty sets, there was no known unbiased sample-based estimator of the above objective prior to our work in the theoretical function approximation literature because of the minimization over $q$; this is in contrast to the particular IPM in~[4], where such an estimator exists and is crucially used in their analysis. This is not an intuitive explanation, it is s statement of fact.
>
>
> **(c)** We do not propose a new algorithm: as stated in the paper, we derive finite-sample and convergence results for an algorithm whose components already existed in the literature. As also stated in the paper,  multi–time-scale algorithms with neural network function approximation (see [5], which we cite) have been studied empirically in the literature. Our contribution is primarily theoretical. We rigorously prove that the two–time-scale stochastic approximation scheme resolves the bias issue arising from the lack of a direct plug-in unbiased single-sample estimator and is sufficient to obtain the claimed finite-time guarantees which are order-wise the same as in the non-robust case. Our work closes the gap between theory and practice in DRRL in this setting.
>
>
> [1] Rayadurgam Srikant and Lei Ying. Finite-time error bounds for linear stochastic approximation and td learning. In Conference on learning theory, pp. 2803–2830. PMLR, 2019.
>
> [2] Jalaj Bhandari, Daniel Russo, and Raghav Singal. A finite time analysis of temporal difference learning with linear function approximation. In Conference on learning theory, pp. 1691–1692. PMLR, 2018.
>
> [3] Zaiwei Chen, John-Paul Clarke, and Siva Theja Maguluri. Target network and truncation overcome the deadly triad in-learning. SIAM Journal on Mathematics of Data Science, 5(4):1078–1101, 2023
>
>
> [4] Kishan Panaganti Badrinath and Dileep Kalathil. Robust reinforcement learning using least squares policy iteration with provable performance guarantees. In International Conference on Machine Learning, pp. 511–520. PMLR, 2021.
>
> [5] Zhipeng Liang, Xiaoteng Ma, Jose Blanchet, Jiheng Zhang, and Zhengyuan Zhou. Single-trajectory distributionally robust reinforcement learning. arXiv preprint arXiv:2301.11721, 2023.

---

### Official Review · Reviewer_XZYB · 2025-11-01

**Soundness:** 3
**Presentation:** 3
**Contribution:** 2
**Rating:** 4
**Confidence:** 4

**Summary:**

The paper provides finite-time convergence guarantees for distributionally robust TD learning with linear function approximation. It analyzes a model-free, two time scale scheme. The paper derives a non-asymptotic error bound showing $O(1/\epsilon^2)$ sample complexity.

**Strengths:**

+ The paper provides a finite sample analysis of distributionally robust policy evaluation with linear approximation.
+ The paper is rigorous and mathematically sound (although I did not check all the proofs) giving finite-time analysis of distributionally robust TD with linear FA.
+ Useful theoretical work for robust RL.

**Weaknesses:**

- The results are restricted to linear function approximation under assumptions, which limits applicability to large-scale or nonlinear deep RL.
- The analysis seem to require very restrictive and strong assumptions, e.g., Assumption 2 requiring that the policy induces an irreducible and aperiodic Markov chain under P0 (hence, mixing assumptions), bounded features, exact projection operator, etc. These conditions are generally unrealistic in RL environments which limits applicability. Also the constants hidden in $\tilde{O}$ may grow poorly with the problem parameters.
- I am not sure what the key technical novelty in analysis is in relation to prior robust TD or adversarial contamination analyses.
- I did not see clear guidance on how to select the ambiguity radius or choose between different uncertainty sets (TV and Wasserstein) from data.
- There is no experimental validation. Some empirical tests on benchmarks would better demonstrate the utility of the bounds.

**Questions:**

1) What specific innovations allow finite time bounds compared to earlier analyses?
2) How tight are the obtained rates? Can you show matching lower bounds?
3) Can the approach extend beyond linear FA?
4) Are the assumptions made in the paper necessary? Can some of the assumptions be relaxed or dispensed with?

---

> ### Author Response · Authors · 2025-11-23
> **Response, Part 1**
>
> We thank the reviewer for the constructive criticism and for providing us with an opportunity to clarify the scope of the paper.
>
>
> **Comment:** The results are restricted to linear function approximation under assumptions, which limits applicability to large-scale or nonlinear deep RL.
>
>
> **Response:** The results are in line with the broader RL theory literature, where most non-robust and robust convergence guarantees are derived for tractable classes such as linear function approximation (see, e.g., [1,2,3,4] for non-robust RL and [5,6,7,8] for robust RL). Though in practice, neural networks are used to approximate the value functions, it remains intractable to provide theoretical convergence guarantees for neural function approximation. Historically, one proves results for linear function and then the ideas were implemented using neural networks. In our case, \cite{liang2023single} has implemented closely-related ideas using neural networks and we have now provided theoretical backing to it by considering linear function approximation. It would be possible to extend the results to neural networks if we use the NTK (neural tangent kernel) theory but that would be along the lines of well-established prior work and would not add much novelty to our paper.
>
> **Comment:** The analysis seem to require very restrictive and strong assumptions, e.g., Assumption 2 requiring that the policy induces an irreducible and aperiodic Markov chain under P0 (hence, mixing assumptions), bounded features, exact projection operator, etc. These conditions are generally unrealistic in RL environments which limits applicability. Also the constants hidden in
>  may grow poorly with the problem parameters.
>
> **Response:**  The only significant assumption we make is that Markov chain is ergodic (Assumption 2). Note that the ultimate goal is to find the optimal policy, which we do using Q-learning. In the case of Q-learning to find the optimal robust policy,
> we just need a behavioral policy $\pi_b$ that satisfies this assumption, This is easy to achieve by choosing a
> behavioral policy that randomizes over all actions at each state.
>
> The boundedness of feature vectors is a historical assumption widely used in RL theory literature with function approximation (starting from the initial theory papers with function approximation (see, [1,2]) to recent works in robust RL ([5]). The Euclidean projection step (Line 7 of the algorithms) is simply a scaling of the dual parameter vector as follows: if the norm of the vector is higher than $B_\nu$ , update it to. $\nu \frac{B_\nu}{\|\nu\|_2}$; else, keep it as is. As $\nu$ lies in a much smaller dimension than the state-action space dimension, this projection step is easily implementable.
>
>
>  Third, regarding the concern about problem dependent constants in the theorem: our analysis shows that the sample complexity upto function approximation erorrs is of order $\mathcal{O}\left(\ln\left(\frac{1}{\epsilon(1-\gamma)}\right)\frac{1}{\epsilon^{2}(1-\gamma)^4}\right)$,
> which is in line with the dependency on the parameter $\gamma$ that is standard in the non-robust RL literature (e.g., [9]).
>
>
>
>
> [1] Rayadurgam Srikant and Lei Ying. Finite-time error bounds for linear stochastic approximation andtd
> learning. In Conference on learning theory, pp. 2803–2830. PMLR, 2019.
>
> [2] Jalaj Bhandari, Daniel Russo, and Raghav Singal. A finite time analysis of temporal difference
> learning with linear function approximation. In Conference on learning theory, pp. 1691–1692.
> PMLR, 2018.
>
> [3] Chi Jin, Zhuoran Yang, Zhaoran Wang, and Michael I Jordan. Provably efficient reinforcement
> learning with linear function approximation. In Conference on learning theory, pp. 2137–2143.
> PMLR, 2020.
>
> [4] Ruosong Wang, Dean P Foster, and Sham M Kakade. What are the statistical limits of offline rl with
> linear function approximation? arXiv preprint arXiv:2010.11895, 2020.
>
> [5] Ruida Zhou, Tao Liu, Min Cheng, Dileep Kalathil, PR Kumar, and Chao Tian. Natural actor-critic
> for robust reinforcement learning with function approximation. Advances in neural information
> processing systems, 36:97–133, 2023.
>
> [6] Kishan Panaganti Badrinath and Dileep Kalathil. Robust reinforcement learning using least squares
> policy iteration with provable performance guarantees. In International Conference on Machine
> Learning, pp. 511–520. PMLR, 2021.
>
> [7] Aviv Tamar, Shie Mannor, and Huan Xu. Scaling up robust mdps using function approximation. In
> International conference on machine learning, pp. 181–189. PMLR, 2014.
>
> [8] Yue Wang and Shaofeng Zou. Online robust reinforcement learning with model uncertainty. Advances
> in Neural Information Processing Systems, 34:7193–7206, 2021.
>
> [9] Zaiwei Chen, John-Paul Clarke, and Siva Theja Maguluri. Target network and truncation overcome
> the deadly triad in-learning. SIAM Journal on Mathematics of Data Science, 5(4):1078–1101,
> 2023

---

> ### Author Response · Authors · 2025-11-23
> **Response, Part 2**
>
> **Comment:** I am not sure what the key technical novelty in analysis is in relation to prior robust TD or adversarial contamination analyses.
>
> **Response:**
> Our work combines the analytical approaches from different domains: target network approach used in non-robust Q-learning to deal with contraction issue, distributionally robust optimization to work in the dual form of the robust RL problem and multi-time scale stochastic approximation to tackle the problem of biased single-sample estimator of the inner optimization problem in DRRL. Each of these analytical techniques and the corresponding convergence have been previously studied in isolation; making these tools work together in the DRRL setting with function approximation is non-trivial. In particular, we need to carefully track their interactions so that the resulting convergence rate yields a sample complexity that is order-wise the same as in the non-robust case, with the same dependencies on $\gamma$.
>
> Here is an example of the non-trivial work that was needed to get the rate of convergence that we were able to obtain. Function approximation in the dual space and the estimate of the dual variables introduce both a bias and a variance in the estimate of the value function. So both have to be controlled appropriately. To study the bias, we use the average of the suboptimality gap in the dual objective at each (state, action) pair weighted by the stationary probability of visiting the (state, action) pair. One could potentially use other metrics to measure the accuracy of the dual variable estimation, but this particular metric is crucial for the proofs to work out. To control the variance, we use averaging of the dual parameter vector (line 9 in the algorithms) in the algorithms, which to the best of our knowledge had not been used previously in this context. Without averaging, the rate of convergence does not match the non-robust case.
>
> **Comment:** I did not see clear guidance on how to select the ambiguity radius or choose between different uncertainty sets (TV and Wasserstein) from data.
>
>
> **Response:**
> We agree that data-driven selection of the ambiguity radius and the uncertainty sets are important practical questions. Our focus in this work is on the theoretical behavior of robust TD/Q-learning under a given DRRL model, and our guarantees hold for any fixed ambiguity radius within that model. To the best of our knowledge, there is no theoretical work on data-driven approaches to the determination of the ambiguity radius and this would certainly be an excellent direction for future work.
>
> **Comment:** There is no experimental validation. Some empirical tests on benchmarks would better demonstrate the utility of the bounds.
>
> **Response:** The contribution of this paper is theoretical. As mentioned in the paper, there have been prior works with empirical validation of robust RL but the theory in those papers are very different from the practical implementations. Our work is the first to address function approximation (without extremely restrictive assumptions) in DRRL.
>
>
> **Comment:** What specific innovations allow finite time bounds compared to earlier analyses?
>
>
> **Response:** Please see the response to the third comment.
>
> **Comment:** How tight are the obtained rates? Can you show matching lower bounds?
>
>
> **Response:** We don't have matching lower bounds. But our finite-time bounds exactly match the state-of-the-art results for the non-robust case in the limit as $\delta\rightarrow 0.$ For non-zero $\delta,$ our finite-time bounds show the same rate of convergence in terms of the number of samples as the non-robust case.
>
> **Comment:** Can the approach extend beyond linear FA?
>
> **Response:**  While the theory is valid for linear FA, the algorithm can be straight-forwardly extended to neural networks to approximate the value functions. The extension to Neural Tangent Kernel (NTK) approximations of neural networks should also be straightforward, but would not be interesting from the point of theoretical novelty.
>
> **Comment:** Are the assumptions made in the paper necessary? Can some of the assumptions be relaxed or dispensed with?
>
>
> **Response:**  Please see the response to the second comment in "Response, Part 1" where we explained why the assumptions are not restrictive when our goal is to learn the optimal policy using Q-learning.

---

### Official Review · Reviewer_bKTe · 2025-11-04

**Soundness:** 1
**Presentation:** 1
**Contribution:** 2
**Rating:** 2
**Confidence:** 4

**Summary:**

This paper addresses temporal-difference (TD) learning with linear function approximation for robust discounted Markov decision processes (MDP) under Total Variation (TV) and Wasserstein uncertainty in transition dynamics. Namely, the authors employ a two tier approach with varied time scales - one to address the inner optimization problem of the dual variables for each state-action pair and another to effectively parameterize the dual variables - which ultimately allow for the derivation of a sample complexity of $\tilde{O}(1/\epsilon^2)$ for uncertainty sets satisfying specific conditions.

**Strengths:**

- The problem being approached is highly relevant and motivated with respect to recent advances in theoretical robust RL.
- The use of a target network mechanism to overcome the projection mismatch arising from linear function approximation to facilitate stable convergence in finite-time enables the algorithm to alleviate the non-contractive nature of the projected robust Bellman operator $\Pi\mathcal{T}^\pi_r$ is fascinating. By having this occur at a "slower" rate, the algorithm can efficiently address the inner optimization problem of finding the worst-case distribution.
- Through the above, the author's were able to derive the final sample complexity in Corollary 1, which aligns with results in the non-robust case. This theoretical work then makes progress on closing the sim-to-real gap.

**Weaknesses:**

- While useful, this work has limited scope. Specifically, in practice one often wants to find some optimal policy $\pi^{*}$, not just evaluating some arbitrary policy, thus limiting the practical significance of the work. The authors briefly discuss a Q-learning extension, however, no formal justification is made, making it's contribution indirect.
- Perhaps more pressing than this is that I believe that there is an error in the proof of Lemma 2. Specifically, Part 3 of Assumption 1 requires an unbiased estimator for the objective function $F(\lambda^a_s)$, meaning that $\mathbb{E}[\sigma]=F(\lambda^a_s).$ The target function is then $F(\lambda^a_s)=\mathbb{E}[\min\{V(X),\lambda^a_s\}]-\delta\lambda^a_s$. On line 10 of Algorithm 1, you use equation 20 to find $\sigma(\cdot;\cdot,\cdot).$ However, by optimizing for $a=\lambda^a_s\in\\{-\frac{1}{1-\gamma},\frac{1}{1-\gamma}\\}$ as in equation 18, equation 20 would only hold if $\lambda^a_s=1$. But from line 122, $0<\gamma<1$ which would imply that the R.H.S of equation 20 should be $\min(V(S'),\lambda^a_s)-\delta\lambda^a_s$. As written currently, we can see that $bias=\mathbb{E}[\sigma]-F(\lambda^a_s)=\delta\lambda^a_s-\delta$. Putting this aside for a moment, how applicable is your algorithm in practice with the underlying assumptions?
- There is not any empirical validation of the proposed algorithm.
- The claim on lines 115-116 is incorrect, see [1].
- Significant number of typos and inconsistent notation. See below for suggestions on actionable edits.

[1] Zachary Roch, George Atia, and Yue Wang. A Reduction Framework for Distributionally Robust Reinforcement Learning under Average Reward. 2025.

**Questions:**

- $\Delta_\mathcal{S}$ on line 121 is not defined. Also, use different notation to mean the same thing on lines 235, 246, and elsewhere.
- Notation for the states and actions are not consistent/clear depending on it's respective use. i.e. use of $s, s', S', S_t$.
- Reuse of $r$ for both the reward function and to denote a robust MDP, robust value function, etc.
- Need citations for the $(s,a)$-rectangularity assumption, i.e. [2,3].
- Increased clarity by showing $\forall s\in\mathcal{S}$ and $\forall(s,a)\in\mathcal{S}\times\mathcal{A}$ when formally writing equations like on line 158 versus equation 50.
- $M$ used in Assumption 1 without definition.
- $X$ used without defining several places, i.e. line 239.
- $d$ used on line 268 before being defined. What is the difference between $\lambda_d$ and $d_\lambda$ as seen on lines 268 and 270, respectively?
- $\mathcal{M}_\nu$ used in equation 6 before being defined.
- Are $\theta^t_k$ and $\theta_{t,k}$ referring to the same thing?
- If $C_{mix}$ is the robust mixing time, it should be formally defined and discussed.
- $c$ is not defined in the algorithm. Similarly $C_e$ on line 352.
- Lemma 1 should come before Theorem 1. There is also not a clear distinction in the wording of these.
- "The noise term $n^\theta_{k+1}$ collects all remaining terms" on line 423 is not precise.
- Typo of "State" instead of "stae" on line 057, line 428, "this", on line 465, "introduction", in the uncertainty set on line 643, at the end of line 953, and a missing space on line 1492.
- Use of $W_\ell$ when discussing TV in the appendix.
- Reuse of notation starting on line 646 where $a\in[m,M].$
- Extra line in the equation on line 799.
- Incomplete sentence on line 819 and on line 1497.
- Is $\lambda(s,a)$ on line 825 the same as $\lambda^a_s$?
- $B_\nu$ is used in the appendix in equation 25 before being defined.
- Period on the wrong line in equation 26 and line 998.
- Does MDS on line 1026 refer to a Martingale Difference Sequence?
- What does the subscript of $\cdot_{op}$ refer to in the notations section in the appendix? Also, the notation section should go before the main proofs and where the notation is subsequently used.
- No clear distinction when a proof ends.

[2] Iyengar, G. N. Robust dynamic programming. 2005.

[3] Nilim, A. and El Ghaoui, L. Robust control of Markov decision processes with uncertain transition matrices.

**Details Of Ethics Concerns:**

Though the author's disclose the use of LLM's to polish the language in certain parts of the paper and further say that all technical content, proofs, and conclusions are the sole work of the authors, I have serious doubts that this was the extent that LLMs were used, though it is difficult to definitively determine. Namely:
- There are an extensive number of typos, reuse of notation, organization, inconsistencies, and non-standard formatting as compared with many theory-based papers. Please see the questions section of my review for an abbreviated list of these action items that I bring to the author's attention.
- While the authors use the notation of $diag(\cdot)$ on line 169, "$operatornamediag(\cdot)$" appears on line 1456.

With the above put together, I suspect that this paper employed LLMs to an extent further than initially disclosed which calls into question the integrity of the underlying work, even if it was pieced together by the authors. The lack of empirical validation, though not necessarily suspicious by itself for a theory-based paper, indicates a higher LLM usage than initially disclosed in my opinion. Therefore, I believe this work may need a closer review.

---

> ### Author Response · Authors · 2025-11-23
> **Response, Part 1**
>
> We thank the reviewer for carefully reading the paper and pointing out typos and inconsistencies in the notations in the paper. We have now revised the paper to fix these issues. We believe that there were no errors in any of the proofs, only typos and notational issues. Hopefully, the revised version makes this fact clear.
>
> **Comment:** While useful, this work has limited scope. Specifically, in practice one often wants to find some optimal policy $\pi^*$, not just evaluating some arbitrary policy, thus limiting the practical significance of the work. The authors briefly discuss a Q-learning extension; however, no formal justification is made, making it's contribution indirect.
>
> **Response:** Indeed, the paper presents a complete algorithm to obtain the oppimal policy. The proof of the Q-learning algorithm is presented in the appendix because the proof is nearly identical to the proof of convergence of robust TD learning. Additionally, from an expository point of view, we felt that the proof of TD learning would be easier for a reader to follow because it avoids the extra max in every step of the algorithm.
>
> **Comment:** Perhaps more pressing than this is that I believe that there is an error in the proof of Lemma 2...
>
> **Response:** We want to clarify that there is no error in the proof. However, there was a typo in equation 20 (line 678) of the original version. The expression for the unbiased estimate of the objective is $min \left(V(S'),\lambda_s^a\right)-\delta \lambda_s^a$.  The term $\lambda_s^a$ was missing. We have fixed the issue in the revised version.
>
> **Comment:** Putting this aside for a moment, how applicable is your algorithm in practice with the underlying assumptions?
>
> **Response:** The ideas closely related to those in our paper, except for the averaging step (line 9 of the algorithm and also equation (8)), have been implemented previously with neural network function approximation(see the paper by [1] which has been cited in the paper, though no theory was proved with any function approximation in [1]). So in that sense, the algorithm is completely implementable since the averaging step is trivial to implement. To the best of our knowledge, there was no proof that the algorithm converged, which is the main point of our paper. Now, regarding the assumptions, the only significant assumption we make is that Markov chain is ergodic. As the reviewer has mentioned, the ultimate goal is to obtain an optimal policy. We do this using Q-learning. In the case of Q-learning, we just need a behavioral policy that satisfies this assumption, This is easy to achieve by choosing a behavioral policy that randomizes over all actions at each state.
>
> **Comment:** There is not any empirical validation of the proposed algorithm.
>
> **Response:** As mentioned in the previous response, the main focus of the paper is a theoretical performance guarantee for the case of function approximation, which did not exist in the prior literature.
>
>
> **Comment:** The claim on lines 115-116 is incorrect, see [2]
>
>
>
> **Response:**
> The paper cited by the reviewer does not use function approximation; it is in the tabular setting. Additionally, it is model-based. So we believe that our statement is correct: "However, to the best of our knowledge, it is worth noting that there are no
> performance guarantees even in the average-reward literature when function approximation is used."
>
> We thank the reviewer for pointing out a recent paper on average reward DRRL, though the statement still holds true because of the above reasons. Moreover, in this paper, we are interested in discounted reward robust RL. Hence, we have not made an exhaustive comparison
> of our work with work on average-reward robust RL. Nevertheless, we have added the citation to the revised version.
>
> **Comment:** Significant number of typos and inconsistent notation. See below for suggestions on actionable edits.
>
> **Response:** We thank the reviewer for pointing out the typos and inconsistencies. In the revised version, we have fixed all the typos and notational inconsistencies pointed out by the reviewer. We checked the paper very carefully to make sure there are no other typos to the best of our knowledge. We improved the overall presentation of the paper and made sure the notations are proper and consistent.
>
>
>
>
> [1] Zhipeng Liang, Xiaoteng Ma, Jose Blanchet, Jiheng Zhang, and Zhengyuan Zhou. Single-trajectory
> distributionally robust reinforcement learning. arXiv preprint arXiv:2301.11721, 2023.
>
> [2] Zachary Roch, George Atia, and Yue Wang. A Reduction Framework for Distributionally Robust Reinforcement Learning under Average Reward. 2025.

---

> ### Author Response · Authors · 2025-11-23
> **Response, Part 2**
>
> **Comment:** Ethics Review Concerns:
>
> **Response:**  We respectfully and strongly protest the comment questioning the integrity of the work. All the technical steps and proofs were solely done by the authors. We did not use an LLM except to polish the writing and the grammar of the paper in some parts. The notational inconsistency and typos in the submitted version are the fault of the authors. We were working on the paper until the last moment and were trying to write the paper in a manner that works for a large class of uncertainty sets. Finalizing the details took a non-trivial amount of time and contributed to the typos. For example, the missing 'backslash' before the 'operatornamediag' in the Appendix is a cut-and-paste mistake since we were working with several versions of the paper. If we had used an LLM for the proofs, we would have stated that we used an LLM as a research assistant since that is an allowed option. We carefully chose the right option based on how we used an LLM.
>
> **Comment:**
> If $C_{mix}$ is the robust mixing time, it should be formally defined and discussed.
>
> **Response:**
> In the revised version we clarify that $C_{\mathrm{mix}}$
>  is the geometric mixing constant for the nominal Markov chain under $P_0$ and policy $\pi$. In the case of Q-learning, it would be simply the mixing time constant associated with the behavioral policy. We do not have a notion of a robust mixing time.

---

### Meta-Review · Area_Chair_727Z · 2026-01-07

**Summary:**

This paper studies temporal-difference (TD) learning method with linear function approximation for robust discounted MDPs, considering uncertainty characterized by Total Variation and Wasserstein ambuguity sets. As the reviewers noted, this paper focuses on policy evaluation for a given fixed policy, rather than the more important problem of policy optimization, which leads to its limited scope. In addition, the theoretical bounds provided include constants that may grow poorly with problem parameters. Based on these limitations, I recommend rejection.

**Reviewer Concerns:**

Some minor comments have been addressed, such as misleading about Lemma 2 caused by typos noted by reviewer bKTe, and the clarification of some claims noted by reviewer FRHC. Concerns stated in summary are still outstanding.

**Reviewer Scores:**

There is no clear evidence to suggest that any reviewer would have changed their score after a full discussion.

---

### Decision · Program_Chairs · 2026-01-26

Reject